# FedLoGe: Joint Local and Generic Federated Learning under Long-tailed Data

**Zikai Xiao**[1][*]**, Zihan Chen**[2][*]**, Liyinglan Liu**[3]**, Yang Feng**[4]**, Jian Wu**[1]**, Wanlu Liu**[1]**,
**Joey Tianyi Zhou**[5,6]**, Howard Hao Yang**[1]**, Zuozhu Liu**[1][†]

[1]Zhejiang University,
[2]Singapore University of Technology and Design,
[3]University of Electronic Science and Technology of China,
[4]Angelalign Technology Inc,
[5]IHPC, Agency for Science, Technology and Research, Singapore,
[6]CFAR, Agency for Science, Technology and Research, Singapore
`zikai@zju.edu.cn`

## Abstract

Federated Long-Tailed Learning (Fed-LT), a paradigm wherein data collected from decentralized local clients manifests a globally prevalent long-tailed distribution, has garnered considerable attention in recent times. In the context of Fed-LT, existing works have predominantly centered on addressing the data imbalance issue to enhance the efficacy of the generic global model while neglecting the performance at the local level. In contrast, conventional Personalized Federated Learning (pFL) techniques are primarily devised to optimize personalized local models under the presumption of a balanced global data distribution. This paper introduces an approach termed **Fed**erated **Lo**cal and **Ge**neric Model Training in Fed-LT (FedLoGe), which enhances both local and generic model performance through the integration of representation learning and classifier alignment within a neural collapse framework. Our investigation reveals the feasibility of employing a shared backbone as a foundational framework for capturing overarching global trends, while concurrently employing individualized classifiers to encapsulate distinct refinements stemming from each client's local features. Building upon this discovery, we establish the Static Sparse Equiangular Tight Frame Classifier (SSE-C), inspired by neural collapse principles that naturally prune extraneous noisy features and foster the acquisition of potent data representations. Furthermore, leveraging insights from imbalance neural collapse's classifier norm patterns, we develop Global and Local Adaptive Feature Realignment (GLA-FR) via an auxiliary global classifier and personalized Euclidean norm transfer to align global features with client preferences. Extensive experimental results on CIFAR-10/100-LT, ImageNet-LT, and iNaturalist demonstrate the advantage of our method over state-of-the-art pFL and Fed-LT approaches. Our codes are available at https://github.com/ZackZikaiXiao/FedLoGe.

## 1 Introduction

Federated learning (FL) enables collaborative model training across decentralized clients without exposing local private data (McMahan et al., 2017; Kairouz et al., 2021). Recent work further investigates the federated long-tailed learning (Fed-LT) task, where the global data exhibits long-tailed distributions and local clients hold heterogeneous distributions (Chen et al., 2022b; Shang et al., 2022b). They usually learn a well-trained generic global model, whose performance might degrade when universally applied to all clients with diverse data, degrading its practical applicability. For example, in the realm of smart healthcare, as demonstrated by Lee & Shin (2020), Chen et al. (2022a),

---

[*]Co-first author.

[†]Corresponding author.

and Elbatel et al. (2023), the capacity of the global model to deliver high-quality diagnostics is limited, as patient distributions vary across specialized hospitals. Additionally, in cross-institutional financial applications like credit scoring (Dastile et al., 2020) and fraud detection (Awoyemi et al., 2017), individuals from different regions or age groups may exhibit dissimilar credit patterns.

A high-quality global model with balanced performance could speed up local adaptation and attract new clients, while personalized models aim to provide enhanced local performance by considering local data characteristics. Nevertheless, existing works on Fed-LT have primarily focused on addressing the imbalance in the context of the global long-tailed data (Yang et al., 2023a; Qian et al., 2023; Xiao et al., 2023), neglecting the tailoring of models to the needs of individual clients, since local data statistics could be diverse and not necessarily long-tailed. Personalized federated learning (pFL) (Tan et al., 2022), which trains customized local models for a single or a group of clients, offers an alternative solution to prioritize each client's (or group's) distinct data statistics and preferences, in which the global generic model is deemed as a bridge for training local personalized models and boosting the local performance with expressive representations (Li et al., 2021b; Collins et al., 2021; Li et al., 2021a). However, conventional pFL approaches are not supposed to attain a superior global generic model in Fed-LT.

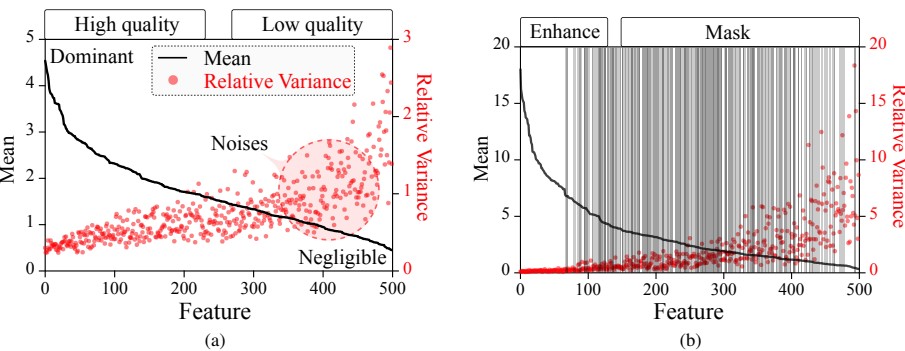

(a)                                    (b)

Figure 1: (a): The mean (sorted in descending order) and variance of class means unveil feature degeneration: The feature collapse property ceases to prevail, and features with diminished means **exhibit substantial variance rather than being zero**; (b): After training the backbone with SSE-C, in which **noisy features with bigger variance are partially pruned** (gray shaded vertical lines), and **enhance the quality** (smaller variance) of dominant features.

Designing a framework to **simultaneously train global and local models** in the presence of Fed-LT remains a critical challenge. Inspired by Kang et al. (2019), we find that the generality and transferability of feature extractors are significantly superior to classifiers (Kim et al., 2022; Vasconcelos et al., 2022). On the contrary, adjusting the classifier has proven to be quite effective in addressing the imbalance and heterogeneity issues (Li et al., 2022a; Zhang et al., 2022a). In other words, the feature extractor can serve as the cornerstone to reflect global trends, while adjusting classifiers can induce the model to adaptively achieve superior personalized performance across the server and heterogeneous clients. Consequently, we conceptualize our model learning with **two intertwined processes: global representation learning and imbalanced/heterogeneous classifier adaptation**. Adopting this viewpoint, we identify two key challenges for personalized Fed-LT:

C1: *How to learn effective representations under heterogeneous and imbalanced data?*

Due to heterogeneity, each client captures different feature distributions, leading to divergence during model aggregation and inferior global performance. Recent studies show that training with a fixed classifier can reduce divergence among heterogeneous clients to improve performance (Oh et al., 2022; Dong et al., 2022). The fixed classifier serves as a consistent criterion for learning representations across clients over time, rather than being an optimal choice itself. For instance, Yang et al. (2022) proposes to initialize the classifier as a simplex equiangular tight frame (ETF) with maximal pairwise angles under imbalanced learning. In general, these fixed classifiers force the feature prototypes to converge to an optimal structure to improve representation learning. However, their effectiveness in resolving Fed-LT regarding both global and local models is not investigated.

We examine the effectiveness of training with fixed classifiers in Fed-LT from the perspective of neural collapse (NC) (Papyan et al., 2020). The NC identifies a salient property of the feature

space, showing that all within-same-class features tend to collapse to their respective class means. However, as shown in Fig. 1(a)[1], preliminary experiments benchmarking ETF with CIFAR-100 in Fed-LT suggest that only a few features have relatively large means, while most of the small-mean features are contaminated by severe noise. Such observations are inconsistent with the feature collapse property, and we coin it as *feature degeneration*. More details of the computation process for the data in Fig. 1 as well as the necessary explanation can be found in Appendix A.2.

To resolve the feature degeneration for improved representation learning, we propose the Static Sparse Equiangular Tight Frame Classifier (SSE-C) inspired by the sparse coding theories (Frankle & Carbin, 2019; Glorot et al., 2011). The assumption of SSE-C is that the small-mean degenerated features contribute little to model performance while forcing sparsity on them would help learn more expressive representations. We refer to the small-mean features as negligible features and the large-mean features as dominant features. The SSE-C dynamically prunes the classifier weights of those small-mean noisy features, while holding more expressive dominant features. Probing into weights trained with SSE-C validates our assumption, as shown in Fig. 1 (b) and experiments.

C2: *How to conduct effective feature realignment to improve the performance of both the generic and personalized models based on data preferences?*

In Fed-LT, the long-tailed global data distribution and heterogeneous local distributions raise the requirements to learn different global and personalized local models for satisfactory performance. Thus, the feature extractor trained with a fixed classifier needs to be realigned to both global and local models. For the global model, it is necessary to realign the model to improve its performance on global tail classes. For the personalized model, the classifier needs to align features with respective heterogeneous local data preferences.

In this work, we unify the feature realignment for both the server and clients under the neural collapse framework. The key idea is to align both global and local classifiers based on the weight norm of the classifiers. Previous works show that classifier weight norms are closely correlated with the corresponding class cardinalities (Kang et al., 2019; Tan et al., 2021; Li et al., 2020b). Further research provides both empirical explanation and theoretical justification that the classifier weight norm is larger for majority classes while smaller for minorities (Kim & Kim, 2020; Dang et al., 2023; Thrampoulidis et al., 2022).

We propose the Global and Local Adaptive Feature realignment (GLA-FR) mod-

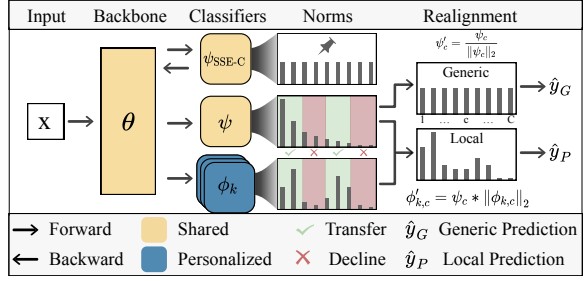

Figure 2: The framework of FedLoGe. The SSE-C enhances the capability of the backbone (feature extractor); during feature alignment, the model transfers the most crucial information from the global classifier to the personalized model, omitting information pertaining to categories with low usage.

ule to align the backbone trained with SSE-C to the server and clients. In particular, we devise auxiliary classifier heads for the global ($\psi$) and $K$ local classifiers ($\{\phi_k\}_{k=1}^K$), which are trained alternately with SSE-C in each epoch, see Algorithm 1. The alignment includes two stages: global alignment and local alignment. The global realignment is simple yet effective, adjusting the weights based on the norms of $\psi$ (Eq.6) to tackle with global balanced test set. The alignment for personalized models is a bit different, as their data distributions differ greatly from the global distribution. We integrate the global trends with each local client's preference by adjusting the global classifiers $\psi$ with the norms of local classifier $\phi_k$ (Eq.7).

Our work represents pioneering efforts to achieve a harmonious integration of global and personalized model learning under Fed-LT, thereby facilitating each participating institution in obtaining a model that is more adeptly tailored to its inherent characteristics and preferences. Comprehensive

---

[1]Our $\psi_{SSE-C}$ leads to two notable improvements in Fig. 1 (b) over Fig. 1 (a): First, it **masks noisy features** with large variances, now in sparse areas marked by grey. Second, the role of **dominant features** is enhanced, as shown by reduced variances among those with larger means, reflecting their increased precision and efficacy.

experiments on representative datasets CIFAR-10/100-LT, ImageNet, and iNaturalist demonstrate the outperformance and efficacy of both global generic and local personalized Fed-LT models.

## 2 RELATED WORK

### 2.1 FEDERATED LEARNING

**Federated Long-tailed Learning** Recent research launched attempts to resolve the Fed-LT task. Model decoupling methods explore frameworks such as classifier retraining (Shang et al., 2022b) and prototype-based classifier rebalancing (Yang et al., 2023a; Dai et al., 2023) for Fed-LT. Shang et al. (2022a) and Wang et al. (2022) investigate calibration and distillation methods to improve the model performance. Many efforts also have been made from the perspective of meta-learning (Qian et al., 2023; Shen et al., 2021), client selection (Zhang et al., 2023; Yang et al., 2021), re-weighting (Wang et al., 2021; Shen et al., 2021), and aggregation (Chou et al., 2022).

**Personalized Federated Learning (pFL)** To deal with the poor generalization performance of the single generic global at local data, pFL has been vastly investigated. A group of works seeks to train local personalized models via transferring knowledge from the generic global model (Li & Wang, 2019; T Dinh et al., 2020; Fallah et al., 2020; Chen et al., 2023). Multi-task learning-based methods are also been explored with client clustering (Sattler et al., 2020; Briggs et al., 2020; Ghosh et al., 2020) and model interpolation (Deng et al., 2020; Li et al., 2021a; Diao et al., 2020).

For neural network-based FL framework, parameter decoupling methods have gained popularity due to their simplicity. Parameter decoupling aims to achieve personalization by decoupling the local private model parameters from the global model parameters. For horizontal decoupling, Li et al. (2021b) personalizes the batch normalization layers, Pillutla et al. (2022) explores decoupling different parts, and personalizing the last layer is adopted in Arivazhagan et al. (2019), Collins et al. (2021), and Briggs et al. (2020). For vertical decoupling, Shen et al. (2022) personalizes channels.

### 2.2 NEURAL COLLAPSE FOR REPRESENTATION LEARNING

Neural collapse refers to a set of four interconnected phenomena that demonstrate a pervasive inductive bias in the terminal phase of training, as shown by Papyan et al. (2020). Subsequently, several works have sought to explain the neural collapse phenomena from the perspective of peeled models (Ji et al., 2021; Fang et al., 2021), unconstrained feature models (Tirer & Bruna, 2022; Mixon et al., 2020; Zhu et al., 2021), and Riemannian manifolds (Yaras et al., 2022). Building upon the findings on neural collapse, Yang et al. (2022) first proposed fixing the classifier to an ETF structure and introduced dot regression loss. The ETF structure was later utilized for semantic segmentation (Zhong et al., 2023), handling heterogeneity in federated learning (Li et al., 2023), transfer learning (Li et al., 2022b), incremental learning (Yang et al., 2023b), and object detection (Ma et al., 2023).

In summary, our method stands out by introducing personalization within the neural collapse framework, effectively overcoming global imbalanced data and enhancing local model personalization through tailored feature distribution alignment. Contrary to FedETF's fixed classifier approach and FedRod's dual classifier structure, our method ensures superior handling of global long-tail bias and precise local model tuning, marking a novel advancement in Fed-LT. For detailed comparisons, please see Section A.13 in the Appendix.

## 3 PROPOSED METHOD

In this section, we introduce our proposed `Fed-LoGe` (see Algorithm 1), a simple yet effective framework to achieve joint personalized and generic model learning for Fed-LT. To boost representation learning and address the global-local inconsistency, we introduce a training paradigm consisting of a sparsified ETF module and global-local feature alignment modules.

### 3.1 PRELIMINARIES

We consider an FL system with $K$ clients and a server. The overall objective is to train $1 + K$ models: 1 generic and $K$ personalized models. Specifically, the generic model is parameterized

by $w = \{\theta, \psi\}$, whereas the $k^{\text{th}}$ personalized model for each client $k \in [K]$ is denoted as $w_k$. We decoupled the neural network models into a feature extractor $f(x, \theta)$ and a set of classifiers. The feature extractor, parameterized by $\theta$, transforms input $x$ into features $h$. The generic classifier $g(h, \psi)$ and personalized classifiers $g(h, \phi_k)$ then map these features to the output labels. The overall global and local objective functions can be respectively expressed as:

$$\textbf{Global:} \min_w \sum_{k=1}^K \frac{|\mathcal{D}_k|}{|\mathcal{D}|} \mathcal{L}_k(w_k, \mathcal{D}_k), \quad \textbf{Local:} \min_{\{\theta, \phi_k\}} \mathcal{L}(\theta, \phi_k; \mathcal{D}_k), \tag{1}$$

where $\mathcal{D} = \{\mathcal{D}_k\}_{k=1}^K$ is the global long-tailed dataset composed of $K$ heterogeneous local datasets, each with size $|\mathcal{D}_k|$. The training is executed over $T$ rounds. In each round $t$, the server distributes the current global model $w^{(t)}$ to all clients for local updates. Furthermore, given that $c$ represents the class index, for $\forall c \in C$, the classifier vector is $\psi_c$, and the corresponding features are $h_c$.

## 3.2 STATIC SPARSE EQUIANGULAR TIGHT FRAME CLASSIFIER (SSE-C)

Initializing the classifier as ETF and subsequently freezing it during training has proven to be an effective strategy in federated learning, attributed to ETF's inherent structure, which ideally exhibits commendable properties of feature collapse under balanced data. However, we found that in the context of Fed-LT, the feature collapse property was not satisfied when initializing the fixed classifiers due to feature degeneracy (the existence of a high-noise feature with a small norm), which is illustrated in Fig. 1 (a). Accordingly, we propose the Static Sparse Equiangular Tight Frame Classifier (SSE-C) via fixing classifier to learn higher-quality features and reduce the impact of negligible features so as to achieve effective representation learning. The server will obtain SSE-C with $\mathcal{L}_{\text{SSE-C}}$ prior to the local training, and all clients will fix SSE-C throughout the training.

We first initialize the classifier as a conventional ETF matrix by

$$\psi = \sqrt{\frac{C}{C-1}} \mathbf{U} \left( \mathbf{I}_C - \frac{1}{C} \mathbf{1}_C \mathbf{1}_C^T \right) \tag{2}$$

where $\psi = [\psi_{:,1}, \cdots, \psi_{:,C}] \in \mathbb{R}^{d \times C}, \mathbf{U} \in \mathbb{R}^{d \times C}$ allows any rotation and satisfies $\mathbf{U}^T \mathbf{U} = \mathbf{I}_C, \mathbf{I}_C$ is the identity matrix. $d$ is the dimension of the classifier vector, and $\mathbf{1}_C$ is an all-ones vector. We can deduce the important property that *all class vector has the equal $\ell_2$ norm and maximal pair-wise angle* $-\frac{1}{C-1}$ *in $\mathbb{R}^d$*. Note that randomly assigning $\beta$ proportion of weights in the ETF matrix to 0 will disrupt the ETF condition: the class vector angles cease to be maximal and equal, and the norms of the classifier vector become unequal. As such, it is necessary to train a sparse ETF structure that satisfies the ETF geometric conditions. We introduce a sparse indicator matrix $\mathbf{S}$ with the same dimensions as $\psi$, where $\beta$ proportion of the elements are randomly set to 0. Then, the sparsified matrix $\psi'$ can be represented as $\psi' = \psi \odot \mathbf{S}$. We design the *Equal Norm Loss* and *Maximal Angle Loss* to optimize the geometric structure to meet the conditions of ETF. First, the $\ell_2$ norm of class vectors should be equal. We constrain all norms of class vector values to a predetermined $\gamma$:

$$l_{\text{norm}}(\psi', \gamma, \mathbf{S}) = \sum_{i=1}^C \left( \left\| \psi'_{:,i} \odot \mathbf{S}_{:,i} \right\|_2 - \gamma \right)^2. \tag{3}$$

Second, we maximize the minimum angle between class vector pairs. Following MMA (Wang et al., 2020), we normalize the classifier vector by $\hat{\psi}'_{:,i} = \frac{\psi'_{:,i}}{\|\psi'_{:,i}\|_2}$ and *maximize only the minimum angle* with the formula:

$$l_{\text{angle}}(\hat{\psi}', \mathbf{S}) = -\frac{1}{C} \sum_{i=1}^C \cos^{-1} \left( \max_{j \in \{1,2,\dots,C\} \setminus \{i\}} \left( (\hat{\psi}'_{:,i} \odot \mathbf{S}_{:,i})^T (\hat{\psi}'_{:,j} \odot \mathbf{S}_{:,j}) \right) \right). \tag{4}$$

By integrating the $l_{\text{norm}}$ and $l_{\text{angle}}$, we obtain the $\mathcal{L}_{\text{SSE-C}}$ for the following training:

$$\mathcal{L}_{\text{SSE-C}} = l_{\text{norm}}(\psi', \gamma, \mathbf{S}) + l_{\text{angle}}(\hat{\psi}', \mathbf{S}). \tag{5}$$

Then we solve the objective $\psi_{\text{SSE-C}} = \arg\min_\psi \mathcal{L}_{\text{SSE-C}}$ by SGD. By regularizing class vector norms and maximizing their minimum angle, the classifier exhibits sparsity while maintaining ETF properties, effectively guiding the model in learning robust features.

---

**Algorithm 1** An overview of `FedLoGe` framework

---

**Input:** $w^{\{0\}} = \{\theta^{(0)}, \psi^{(0)}\}, \{\phi_k^{(0)}\}_{k=1}^K, K, E, T, C$
**Output:** $w^{\{T\}} = \{\theta^{(T)}, \psi'^{(T)}\}, \{w_k\}_{k=1}^K = \{\theta^{(T)}, \phi_k'^{(T)}\}_{k=1}^K$

    **Stage 1: Representation Learning with SSE-C**
1: Obtain $\psi_{\text{SSE-C}}$ by Equation 5
2: **for** each round $t = 1$ **to** $T$ **do**
3:    $S_t \leftarrow$ subset of selected clients
4:    **for** each client $k \in S_t$ **in parallel do**
5:       $\theta_k^t, \psi_k^t \leftarrow \text{CLIENTUPDATE}(k, \theta^{(t-1)}, \psi^{(t-1)}, \psi_{\text{SSE-C}})$
6:       $\theta^{(t)}, \psi^{(t)} = \text{AGGREGATION}(\theta_k^{(t-1)}, \psi_k^{(t-1)})$, for all $k \in S_t$

    **Stage 2: Global Feature Realignment**
7: **for** each classifier vector $c = 1$ **to** $C$ **do**
8:    $\psi_c'^{(T)} \leftarrow \psi_c^{(T)} / \left\|\psi_c^{(T)}\right\|_2$ (Equation 6)     \\ *Align the long-tailed norm to balanced norm*

    **Stage 3: Local Feature Realignment**
9: **for** each client $k = 1$ **to** $K$ **do**
10:    **for** each classifier vector $c = 1$ **to** $C$ **do**
11:       $\phi_{k,c}'^{(T)} \leftarrow \psi_c^{(T)} * \left\|\phi_{k,c}^{(T)}\right\|_2$ (Equation 7) \\ *Incorporate Global Classifier with local statistics*
12:       $\phi_k'^{(T)} \leftarrow \phi_k'^{(T)} - \eta\nabla_\phi(\mathcal{L}(\theta^T, \phi_k'^{(T)}); x)$    \\ *Finetune $\phi_k'^{(T)}$*
13: **return** $w^{(T)} = \{\theta^{(T)}, \psi^{(T)}\}, \{w_k\}_{k=1}^K = \{\theta^{(T)}, \phi_k^{(T)}\}_{k=1}^K$

    **function**CLIENTUPDATE($k, \theta^{(t)}, \psi^{(t)}, \psi_{\text{SSE-C}}$)
1: $\theta_k^{(t)}, \psi_k^{(t)} = \theta^{(t)}, \psi^{(t)}$
2: **for** each local epoch $i = 1$ **to** $E$ **do**
3:    Compute features $h_i \leftarrow f(x_i, \theta^{(t)})$
4:    $\theta_k^{(t+1)} \leftarrow \theta_k^{(t)} - \eta\nabla_\theta(\mathcal{L}(\psi_{\text{SSE-C}}; x_i))$       \\ *Fix $\psi_k^{(t)}, \phi_k^{(t)}$, Update $\theta_k^{(t)}$ with $\psi_{\text{SSE-C}}$*
5:    $\left.\begin{array}{l}\psi_k^{(t+1)} \leftarrow \psi_k^{(t)} - \eta\nabla_\psi(\mathcal{L}(\theta^t, \psi_k^{(t)}); h_i) \\ \phi_k^{(t+1)} \leftarrow \phi_k^{(t)} - \eta\nabla_\phi(\mathcal{L}(\theta^t, \phi_k^{(t)}); h_i)\end{array}\right\}$ \\ *Fix $\theta_k^{(t)}$, Update $\psi_k^{(t)}, \phi_k^{(t)}$*
6: **return** $\theta_k^{(t+1)}, \psi_k^{(t+1)}$

---

### 3.3 GLOBAL AND LOCAL ADAPTIVE FEATURE REALIGNMENT (GLA-FR)

To be adapted in both global and local models, the feature extractor trained with a fixed classifier needs to be realigned to address the imbalance and heterogeneity. Hence, we conduct feature realignment to both global and personalized models after training the SSE-C guided feature extractor, where the realignment should be consistent with the local data statistics/class cardinality.

To obtain a good estimation of class cardinality, in prior work such as Kang et al. (2019); Tan et al. (2021); Li et al. (2020b), the classifier weight norms $\|\psi_c\|$ are found to be correlated with the corresponding class cardinalities $n_c$, in which $\psi_c$ is the classifier weight vector for the $c$-th class. Kim & Kim (2020) provides an explanation from the perspective of decision boundaries - the weight vector norm for more frequent classes is larger, biasing the decision boundary towards less frequent classes. Also, for the neural collapse framework with imbalanced data, the relations between the weight norm of classifiers and the class cardinality also exist (Thrampoulidis et al., 2022; Dang et al., 2023). These findings are consistent, which motivates us to measure/estimate local data statistics based on the norm weight of the classifier.

The frozen ETF classifier is not suitable for feature alignment, attributed to the lack of valid norms to estimate class cardinality. We design a new auxiliary global head $\psi$ to obtain valid norms, which participate in gradient updates and weight aggregation alongside the backbone. After $T$ rounds of training, we get the global weight $w^{(T)} = \{\theta^{(T)}, \psi^{(T)}\}$. The $\theta^{(T)}$ is well trained with $\psi_{SSE-C}$.

For the global adaptive feature distribution process (GA-FR), let $\psi_c$ denote a classifier vector corresponding to the $c^{\text{th}}$ class, where $c \in C$. The aligned classifier vector $\psi_c'$ can be obtained by dividing

| Dataset | Non-IID | α = 0.5 | | | | α = 1 | | | |
| | Imbalance Factor | IF=50 | | IF=100 | | IF=50 | | IF=100 | |
| | Method/Model | GM | PM | GM | PM | GM | PM | GM | PM |
|---|---|---|---|---|---|---|---|---|---|
| CIFAR-10-LT | FedAvg | 0.7988 | 0.8722 | 0.7214 | 0.8664 | 0.7890 | 0.8916 | 0.7231 | 0.8935 |
| | FedProx | 0.7869 | 0.8653 | 0.7160 | 0.8617 | 0.7790 | 0.8888 | 0.7266 | 0.8897 |
| | FedBN | 0.7604 | 0.8837 | 0.6984 | 0.8847 | 0.7596 | 0.8916 | 0.7033 | 0.8925 |
| | FedPer | - | 0.8935 | - | 0.8931 | - | 0.8918 | - | 0.8999 |
| | FedRep | 0.7938 | 0.8988 | 0.7218 | 0.8984 | 0.7816 | 0.9043 | 0.7271 | 0.9065 |
| | Ditto | 0.7813 | 0.8926 | 0.7180 | 0.8874 | 0.7804 | 0.8967 | 0.7210 | 0.8973 |
| | FedROD | 0.7862 | 0.9030 | 0.7137 | 0.8941 | 0.7776 | 0.9025 | 0.7236 | 0.9041 |
| | FedBABU | 0.7851 | 0.8664 | 0.7280 | 0.8613 | 0.7819 | 0.8914 | 0.7316 | 0.8898 |
| | FedETF | 0.8056 | 0.7446 | 0.6709 | 0.7323 | 0.7453 | 0.8106 | 0.6615 | 0.7917 |
| | Ratio Loss | 0.7995 | 0.8791 | 0.7290 | 0.8724 | 0.7857 | 0.8938 | 0.7342 | 0.8934 |
| | FedLoGe | **0.8277** | **0.9112** | **0.7672** | **0.9096** | **0.8189** | **0.9104** | **0.7593** | **0.9073** |
| CIFAR-100-LT | FedAvg | 0.4271 | 0.6019 | 0.3818 | 0.6214 | 0.4215 | 0.6086 | 0.3797 | 0.6269 |
| | FedProx | 0.4276 | 0.6071 | 0.3856 | 0.6214 | 0.4187 | 0.6089 | 0.3788 | 0.6261 |
| | FedBN | 0.4332 | 0.6657 | 0.3895 | 0.6703 | 0.4229 | 0.6373 | 0.3847 | 0.6562 |
| | FedPer | - | 0.6888 | - | 0.7007 | - | 0.6476 | - | 0.6737 |
| | FedRep | 0.4355 | 0.6911 | 0.3907 | 0.6917 | 0.4283 | 0.6539 | 0.3828 | 0.6831 |
| | Ditto | 0.4312 | 0.6441 | 0.3847 | 0.6588 | 0.4188 | 0.6182 | 0.3822 | 0.6320 |
| | FedROD | 0.4384 | 0.7124 | 0.3919 | 0.6919 | 0.4289 | 0.6667 | 0.3902 | 0.6917 |
| | FedBABU | 0.4415 | 0.6416 | 0.3921 | 0.6480 | 0.4336 | 0.6354 | 0.3907 | 0.6598 |
| | FedETF | 0.4223 | 0.6055 | 0.3825 | 0.6421 | 0.4278 | 0.6302 | 0.4278 | 0.6507 |
| | Ratio Loss | 0.4326 | 0.6152 | 0.3912 | 0.6348 | 0.4253 | 0.6213 | 0.3839 | 0.6344 |
| | FedLoGe | **0.4762** | **0.7229** | **0.4233** | **0.7285** | **0.4860** | **0.7099** | **0.4330** | **0.7195** |

Table 1: Test accuracies of our and SOTA methods on CIFAR-10/100-LT with diverse imbalanced and heterogeneous data settings. GM/PM denotes Global/Personalized model.

$\psi_c$ by its $\ell_2$ norm as follows:

$$\psi'_c = \psi_c / \|\psi_c\|_2 \tag{6}$$

Here, $\|\psi_c\|_2$ represents the $\ell_2$ norm of $\psi_c$. Each $\psi'_c$ will be a unit vector, preserving the direction of $\psi_c$ and possessing the magnitude of 1.

For personalized adaptive feature realignment (LA-FR), we adapt the global auxiliary classifier $\psi$ to the personalized classifier by multiplying the norm of $\phi_k$, which implies that clients will leverage information from categories with a larger sample size while omitting information pertaining to the rare categories. For local classifier vector $\phi_{k,c}$ at client $k$, the process of LA-FR is as follows:

$$\phi'_{k,c} = \psi_c * \|\phi_{k,c}\|_2 \tag{7}$$

### 3.4 ALGORITHMS

Overall, our framework `Fed-LoGe` consists of three critical stages: representation learning with SSE-C, global feature realignment, and local feature realignment, for the training of the shared backbone $\theta$, global auxiliary classifier $\psi$, and $K$ local classifiers ($\{\phi_k\}_{k=1}^K$), respectively.

In the first stage, the server first constructs the SSE-C with Eq. 5 before the training and then distributes it to all clients. Upon receiving SSE-C, each client fixes it as the classifier to train the backbone $\theta$, global classifier $\psi$, and local classifier $\phi_k$ alternately. Specifically, we update $\theta$ with fixed $\psi_{\text{SSE-C}}$. Subsequently, the $\theta$ is frozen to update the global head $\psi$ and each local classifier $\phi_k$. At the end of each round, the $\theta$ and $\psi$ are aggregated at the server, while $\phi_k$ is retained locally.

Global adaptive feature realignment (GA-FR) is performed in the second stage, where each class vector is redistributed by the server according to its individual norms, as outlined by Eq. 6. Subsequently, in the third phase, personalized adaptive feature realignment (LA-FR) for the class vectors of the global auxiliary head $\psi$ is performed. Following LA-FR, local finetuning could be further conducted to boost the model performance. A summary of `Fed-LoGe` is given in Algorithm 1.

## 4 EXPERIMENTS

### 4.1 EXPERIMENTAL SETUP

**Dataset, Models and Metrics:** We consider image classification tasks for performance evaluation on benchmark long-tailed datasets: CIFAR-10/100-LT, ImageNet-LT, and iNaturalist-User-

| Dataset | ImageNet | | | | | Inaturalist | | | | |
|---|---|---|---|---|---|---|---|---|---|---|
| Method/Model | Many | Med | Few | GM | PM | Many | Med | Few | GM | PM |
| FedAvg | 0.481 | 0.307 | 0.159 | 0.329 | 0.528 | 0.591 | 0.418 | 0.238 | 0.425 | 0.590 |
| FedProx | 0.493 | 0.318 | 0.180 | 0.343 | 0.500 | 0.525 | 0.484 | 0.223 | 0.432 | 0.596 |
| FedBN | 0.471 | 0.300 | 0.168 | 0.319 | 0.504 | 0.573 | 0.396 | 0.221 | 0.413 | 0.563 |
| FedPer | - | - | - | | 0.653 | - | - | - | - | 0.638 |
| FedRep | 0.460 | 0.309 | 0.187 | 0.330 | 0.574 | 0.571 | 0.453 | 0.237 | 0.429 | 0.627 |
| Ditto | 0.492 | 0.319 | 0.176 | 0.342 | 0.674 | **0.598** | 0.452 | 0.243 | 0.437 | 0.584 |
| FedROD | 0.483 | 0.305 | 0.165 | 0.331 | 0.7033 | 0.585 | 0.416 | 0.243 | 0.421 | 0.699 |
| FedBABU | 0.443 | 0.240 | 0.055 | 0.230 | 0.425 | 0.561 | 0.401 | 0.199 | 0.377 | 0.696 |
| FedETF | 0.425 | 0.239 | 0.05 | 0.222 | 0.418 | 0.587 | 0.431 | 0.245 | 0.437 | 0.713 |
| Ratio Loss | **0.495** | 0.337 | 0.189 | 0.351 | 0.521 | 0.587 | 0.454 | 0.290 | 0.452 | 0.589 |
| FedLoGe | 0.430 | **0.373** | **0.285** | **0.356** | **0.726** | 0.519 | **0.508** | **0.473** | **0.503** | **0.759** |

Table 2: Test accuracies of our and SOTA methods on ImageNet-LT and iNaturalist-160k with diverse heterogeneous data settings.

160k (Van Horn et al., 2018). The CIFAR-10/100-LT datasets are sampled into a long-tailed distribution employing an exponential distribution governed by the Imbalance Factor (IF) in Cao et al. (2019). All experiments are conducted with non-IID data partitions, implemented by the Dirichlet distributions-based approach with parameter $\alpha$ to control the non-IIDness (Chen & Chao, 2022).ResNet-18 is trained over $K = 40$ clients on CIFAR-10-LT, while ResNet-34 and ResNet-50 are implemented on CIFAR-100-LT and ImageNet-LT, respectively, with $K = 20$ clients. The configurations for iNaturalist-160k align with those utilized for ImageNet-LT. We use $\alpha = 1, 0.5$ and IF = 50, 100 in CIFAR-10/100-LT. We use $\alpha = 0.1, 0.5$ in ImageNet-LT and iNaturalist, respectively.

A global balanced dataset is used for the calculation of test accuracy to evaluate global model (GM) performance. We also report the accuracy across many, medium, and few classes. The detailed categorization for many/med/few classes can be found in the Appendix.

For personalized model (PM) evaluation, we use local test accuracy, and the local test set is sampled from the global test set. Each local test set has an identical distribution to the local training set. The accuracy of the PM is the arithmetic mean of local test accuracy across all clients.

**Compared Methods:** In addition to `FedAvg` and `FedProx` (Li et al., 2020a) which are included for reference, we consider two types of state-of-the-art baselines: (1) pFL methods, including `FedBN` (Li et al., 2021b), `FedPer` (Arivazhagan et al., 2019), `FedRep` (Collins et al., 2021), `Ditto` (Li et al., 2021a), and `FedROD` (Chen & Chao, 2022). (2) Federated (Long-tailed) Representation learning, including `FedBABU` (Oh et al., 2022), `FedETF` (Li et al., 2023) and `Ratio Loss` (Wang et al., 2021).

| ETF | SSE-C | GA-FR | LA-FR | GM | PM |
|---|---|---|---|---|---|
| ✓ | | | | 0.3825 | 0.6421 |
| | ✓ | | | 0.4175 | 0.6865 |
| | ✓ | ✓ | | **0.4350** | 0.6868 |
| | ✓ | | ✓ | 0.4175 | 0.7339 |
| ✓ | | ✓ | ✓ | 0.3988 | 0.7043 |
| | ✓ | ✓ | ✓ | **0.4350** | **0.7343** |

Table 3: Ablations of SSE-C and GLA-FR.

## 4.2 PERFORMANCE COMPARISON

For evaluation on CIFAR-10-LT and CIFAR-100-LT, `FedLoGe` consistently outperforms all baselines over all settings, achieving the highest overall accuracies in all settings for both GM and PMs; see Tab. 1. For ImageNet-LT and iNaturalist-160k, Tab. 2 highlights that `FedLoGe` consistently surpasses all baselines, marking significant accuracy improvements, particularly in middle and tail classes. Overall, benefiting from the enhanced representation learning and classifier realignments, `FedLoGe` attains superior GM performance, attributed to the SSE-C design for representation learning, and simultaneously obtains impressive PM performance together with refined feature alignment.

## 4.3 ABLATION STUDY AND SENSITIVITY ANALYSIS

**Ablations of SSE-C and GLA-FR:** In the ablation study, we evaluate the SSE-C and GA-FR/LA-FR individual impacts with respect to both GM and PM performance, over CIFAR-100-LT (IF=100, $\alpha = 0.5$), as given in Tab. 5. The results lead to the following conclusions: Compared to dense ETF, SSE-C can train a superior backbone, enhancing both GM and PM performance. GA-FR

can be combined with any backbone to boost GM performance, while LA-FR can be used with any backbone to boost PM performance. Employing SSE-C and GLA-FR simultaneously yields significant enhancements in both GM and PM performance.

**Negligible and dominant features with SSE-C:** We investigated the effects of pruning on different features over CIFAR-100-LT ($\alpha$=0.5, IF=100). We computed the class mean for each category. Within each class mean, we ordered the means and pruned the corresponding classifier vector weights in descending (from dominant features) and ascending order (from negligible features) with pruning ratios from 0 to 100%. After training with SSE-C, pruning dominant features drastically decreases performance, while pruning negligible features barely affects performance, which is visualized in Fig. 3 (a, b). This observation indicates that SSE-C can adeptly learn more effective features while autonomously disregarding high-noise features.

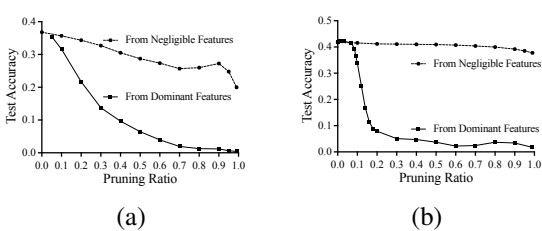

(a)         (b)

Figure 3: (a): Pruning features from smaller means and larger means, respectively. Negligible features exert a minor impact on model performance; (b): Pruning Experiments with SSE-C. The model optimally enhances the dominant features, rendering the impact of negligible features imperceptible.

**Sensitivity analysis for $\gamma$ in SSE-C:** We evaluate the model performance with various values of $\gamma$ (which is assigned as the norm of SSE-C in **norm equal loss**) over CIFAR-100-LT (IF=100, $\alpha = 0.5$), as depicted in Fig. 4 (a). Our observations indicate that a smaller norm leads to a reduced gradient during backpropagation, producing subpar performance. Conversely, a larger norm enables faster convergence. For CIFAR-10/100-LT, the optimal value is approximately 0.1, whereas, for ImageNet and iNaturalist, it is approximately 1.6.

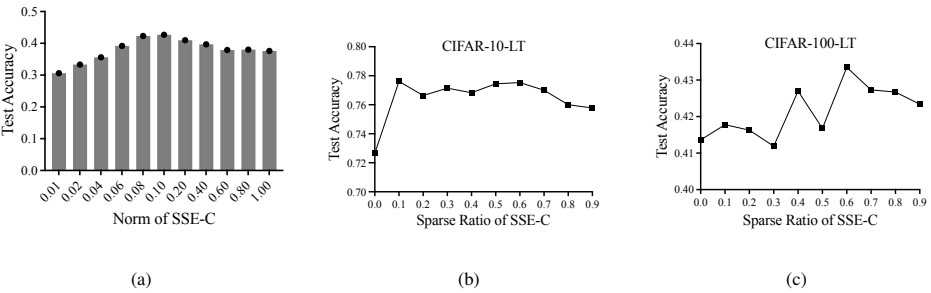

(a)        (b)        (c)

Figure 4: (a): The impact of varying norms $\gamma$ in SSE-C on model performance within CIFAR-100-LT; (b) The impact of sparse ratio $\beta$ on model performance in CIFAR-10-LT; (c): The impact of sparse ratio $\beta$ on model performance in CIFAR-100-LT.

**Sensitivity analysis for Sparse Ratio $\beta$ in SSE-C:** The $\beta$ indicates the pruning proportion in SSE-C. We evaluate the performance with the sparsity from 0 to 90% at intervals of 10% on CIFAR10/100-LT (IF = 100, $\alpha = 0.5$). As shown in Fig. 4 (b) and (c), minor sparsification yields a slight model performance enhancement, and the performance is obtained around 60% sparsity. Surprisingly, a large sparsity ratio still remains superior compared to models without sparsification.

## 5 CONCLUSION

This paper presented `FedLoGe`, a model training framework that enhances the performance of both local and generic models in Fed-LT settings in the unified perspective of neural collapse. The proposed framework is comprised of SSE-C, a component developed inspired by the feature collapse phenomenon to enhance representation learning, and GLA-FR, which enables fast adaptive feature realignment for both global and local models. As a result, `FedLoGe` attains significant performance gains over current methods in personalized and long-tail federated learning. Future research will explore adaptive sparsity and expand the framework to diverse loss functions and tasks.

ACKNOWLEDGEMENTS

This work is supported by the National Natural Science Foundation of China (Grant No. 62106222, No. 62201504), the Natural Science Foundation of Zhejiang Province, China(Grant No. LZ23F020008, No. LGJ22F010001), Zhejiang Lab Open Research Project (No. K2022PD0AB05) and the Zhejiang University-Angelalign Inc. R&D Center for Intelligent Healthcare.

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

## A APPENDIX

### A.1 DETAILED SETUP

**Dataset, Models and Metrics:** We consider image classification tasks for performance evaluation on benchmark long-tailed datasets: CIFAR-10/100-LT, ImageNet-LT, and iNaturalist-User-160k. The CIFAR-10/100-LT datasets are sampled into a long-tailed distribution employing an exponential distribution governed by Imbalance Factor (IF) in Cao et al. (2019), calculated over the comprehensive dataset $\mathcal{D}$. Consistency is maintained by utilizing configurations from Liu et al. (2019) for ImageNet-LT, where the number of images per class varies between 5 and 1280. Additionally, to evaluate performance with real-world data, the study includes experiments on iNaturalist-User-160k, comprising 160k examples across 1023 species classes, sampled from iNaturalist-2017 in Van Horn et al. (2018).

All experiments employ non-IID data partitions, achieved through Dirichlet distributions. The concentration parameter $\alpha$ serves to regulate the identicalness of local data distributions among all clients. A ResNet-18 is trained over $K = 40$ clients on CIFAR-10-LT, while ResNet-34 and ResNet-50 are implemented on CIFAR-100-LT and ImageNet-LT, respectively, with $N = 20$ clients. The configurations for iNaturalist-160k align with those utilized for ImageNet-LT.

The balanced dataset is utilized for the evaluation of global model performance. The global accuracy is calculated as the ratio of the number of correct predictions to the total number of predictions. We also report the accuracy across many, medium, and few classes. The categorization process unfolds as follows: Initially, all classes are sorted based on the number of data samples in descending order. Subsequently, two thresholds (e.g., $75\%$, $95\%$) are set to segregate the classes into head classes, middle classes, and tail classes.

For CIFAR10/100-LT datasets, we engaged 40 clients with full participation in each training round, setting the number of local epochs to 5. For ImageNet/iNaturalist, the experiments involved 20 clients, with a $40\%$ participation rate in each training round, and the number of local epochs was set to 3.

When assessing personalized models, the test set is sampled from the global balanced dataset by local distribution. We ensure that the distribution of the local train set and the local test set is perfectly aligned. The accuracy of local models is the arithmetic mean of all clients.

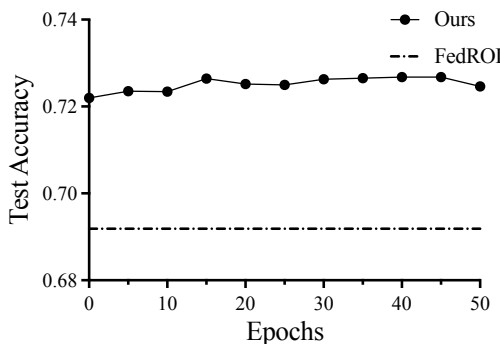

Figure 5: Accuracy of local training after GLA-FR.

**Training Settings:** We train the generic model with $T = 500$ rounds via applying SGD optimizer for local training in all experiments unless otherwise stated. We use $\alpha = 1, 0.5$ and IF = $50, 100$ in CIFAR-10/100-LT. We use $\alpha = 0.1, 0.5$ in ImageNet-LT and iNaturalist, respectively.

## A.2 Table of Notations

Please refer to Table 4 for the table of notations used throughout this paper.

## A.3 Detailes of feature degeneration

Given that $n_c$ is the sample number of class $c$. $\mu_c$ is the class mean of class $c$, and $h_{i,c}$ is the feature of class $c$ on sample $i$, the features $h_{i,c}$ will collapse to the within-class mean $\mu_c = \frac{1}{n_c} \sum_{i=1}^{n_c} h_{i,c}$. This indicates that the covariance $\frac{1}{n_c} \sum_{i=1}^{n_c} (h_{i,c} - \mu_c)(h_{i,c} - \mu_c)^T$ will converge to 0. We investigate whether $h_{i,c} - \mu_c$ tends toward zero when training with a fixed ETF classifier, as shown in Fig. 1. The steps to obtain the data in Fig. 1 are: 1. Select class $c$. 2. During global test, compute all sample features $h_{i,c}$, representing the number of features. Use $h_{i,c}[d]$ to denote the $d^{th}$ feature in $h_{i,c}$. 3. Compute the mean and relative variance of $h_c$, sorting them in descending order of the mean, and accordingly adjust the variance positions. It becomes evident that not all features, $h_{i,c}[d] - \mu_c[d]$, converge to zero, and smaller $\mu_c[d]$ values exhibit greater convergence noise. The process illustrated in Fig. 1 (b) follows the same steps as Fig. 1 (a), but with grey vertical shadows for the sparsely positioned features. It can be observed that sparse ETF training effectively squeezes poor-quality and noisy features into these sparsely allocated positions.

## A.4 Computational Cost

We evaluate the computational expense of `Fed-LoGe` across its three stages. In the initial representation stage, the server is tasked with constructing the SSE-C. Setting the learning rate at 0.0001 and executing 10,000 optimization steps, we conducted the experiments three times on the PyTorch platform utilizing the NVIDIA GeForce RTX 3090; the average cost is 11 minutes and 52 seconds.

| Notation | Description |
|---|---|
| $K$ | Number of clients in the federated learning system. |
| $w$ | Parameters of the generic model, consisting of $\theta$ and $\psi$. |
| $w_k$ | Parameters of the $k^{\text{th}}$ personalized model for client $k$. |
| $f(x, \theta)$ | Feature extractor parameterized by $\theta$. |
| $h$ | Features transformed from input $x$ by feature extractor $f$. |
| $g(h, \psi)$ | Generic classifier mapping features $h$ to output labels. |
| $g(h, \phi_k)$ | Personalized classifier for client $k$. |
| $\phi_k$ | Personalized classifier parameters for client $k$. |
| $\mathcal{D}$ | Global long-tailed dataset. |
| $\mathcal{D}_k$ | Local dataset of client $k$. |
| $T$ | Number of training rounds. |
| $c$ | Class index. |
| $\psi_c$ | Classifier vector corresponding to class $c$. |
| $h_c$ | Features corresponding to class $c$. |
| $\mathbf{S}$ | Sparse indicator matrix. |
| $\gamma$ | Predetermined $\ell_2$ norm value for class vectors. |
| $\mathcal{L}_{\text{SSE-C}}$ | Loss function for Static Sparse Equiangular Tight Frame Classifier (SSE-C). |
| $\mathcal{L}_{\text{norm}}, \mathcal{L}_{\text{angle}}$ | Loss functions for optimizing the geometric structure of SSE-C. |
| $n_c$ | Sample number of class $c$. |
| $\mu_c$ | Class mean of features for class $c$. |
| $h_{i,c}$ | Feature of class $c$ for sample $i$. |

Table 4: Summary of notations used in the paper.

| Method | Time Cost/Round |
|---|---|
| FedAvg | 3min11s |
| FedProx | 5min24s |
| FedRep | 4min07s |
| FedBABU | 3min14s |
| FedLoGe | 3min19s |

Table 5: Computational cost of Fed-LoGe.

## A.5  LOCAL FINETUNE FURTHER IMPROVE PERFORMANCE

After the personalized feature realignment, we proceed with local training without any aggregation. On CIFAR100-LT with IF $= 100$ and $\alpha = 0.5$, the accuracy consistently increases with additional local epochs, reaching stability at 15 epochs, as illustrated in Fig. 5.

## A.6  T-SNE VISUALIZATION OF CLASS MEANS

In this study, we compare features from models trained with SSE-C (sparse) and ETF (dense) using t-SNE, a common method for reducing dimensions, shown in Fig. 6. This helps us see how different training methods affect feature distribution. We notice that with SSE-C, the average features of each class are evenly spread out and quite similar in distance to each other. On the other hand, the model trained with ETF has many class averages very close or even overlapping, making the overall spread more clustered and the model more likely to make mistakes.

## A.7  NORM AND ANGLES OF SSE-C

We have individually computed the statistical measures (mean of norm, variance of norm, mean of mutual angles, and variance of mutual angles) of the classifier vectors within SSE-C under varying

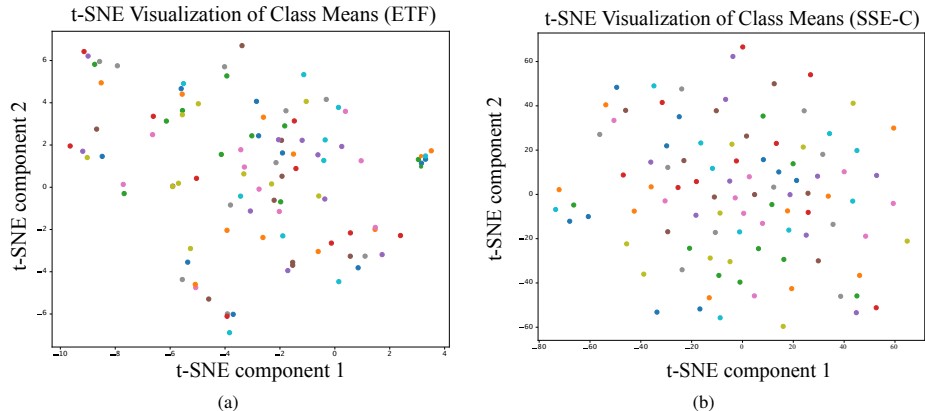

Figure 6: t-SNE visualization of class means of ETF (dense) and SSE-C (sparse)

| Sparse Ratio | Norm | Mean of Norm | Var of Norm | Mean of Angle | Var of Angle |
|---|---|---|---|---|---|
| 0.6 | 0.5 | 0.50 | 3.55e-12 | 90.06 | 0.09 |
| 0.6 | 1.0 | 1.00 | 4.75e-11 | 90.06 | 0.66 |
| 0.6 | 1.5 | 1.50 | 1.71e-11 | 90.04 | 1.21 |
| 0.4 | 0.5 | 0.50 | 1.10e-11 | 90.06 | 0.03 |
| 0.4 | 1.0 | 1.00 | 7.14e-12 | 90.06 | 0.15 |
| 0.4 | 1.5 | 1.50 | 3.5e-11 | 90.06 | 0.79 |

Table 6: Construct measures of SSE-C

| Model | PM(Avg) | PM(Macro) | PM(Weighted) |
|---|---|---|---|
| LA-LR (real distribution) | 0.6200 | 0.2680 | 0.5604 |
| LA-LR (norm) | 0.7273 | 0.5316 | 0.7129 |

Table 7: Local adaptive feature alignment of real distribution and personalized norm

sparse ratios $\beta$ and assigned norms $\gamma$, as shown in Table 6. It is observable that the constructed SSE-C nearly fulfills the properties of an Equiangular Tight Frame (ETF): equal norms, and maximized, equal mutual angles. Different sparse ratios and assigned norms do not significantly influence the final construction.

## A.8 LOCAL NORM OR LOCAL DISTRIBUTION FOR LA-FR

Upon the backbone $\theta$, trained through $\psi_{SSE-C}$, we proceed with local adaptive feature realignment. In our methodology, local norm is employed to align features with personalized preference. It is intuitive to consider that the real local distribution is more representative of the model's preference, and moreover, the local distribution is free for utilization without any computation. Consequently, we explore whether local norm or local distribution yields superior performance.

On CIFAR100-LT with $\alpha = 0.5$ and IF $= 100$, we realign personalized models employing both norm and distribution, calculating the arithmetic mean, macro average, and weighted average of all clients' performance. The results can be seen in Tab. 7. We discern that utilizing norm for realignment is more effective than employing the real distribution. One rationale is that the norm more aptly signifies the model's cognitive capacity over the dataset. For instance, although class A encompasses more samples than class B, B is more prone to misprediction owing to its high resemblance with other classes. In this scenario, realignment ought to adhere to model cognition, fortifying class A over B.

## A.9 PROJECTION LAYER AND DOT-REGRESSION LOSS IS NOT NECESSARY FOR FED-LOGE

In previous work, both the projection layer and dot-regression loss are deemed essential for training the backbone (Li et al. (2023); Yang et al. (2023b)). The projection layer, being a dense structure,

| Method | Many | Med | Few | All |
|---|---|---|---|---|
| ETF | 0.6904 | 0.4751 | 0.1820 | 0.3825 |
| ETF+DR Loss | 0.6882 | 0.4607 | 0.2059 | 0.3932 |
| ETF+Proj+DR Loss | 0.6909 | 0.5011 | 0.2310 | 0.4109 |
| ETF+Proj+DR Loss+Sparse | 0.7013 | 0.5142 | 0.2479 | 0.4237 |
| FedLoGe | 0.7137 | 0.4989 | 0.2179 | 0.4249 |
| FedLoGe+Proj | 0.7014 | 0.4642 | 0.1887 | 0.4004 |

Table 8: Ablations of projection layer and dot-regression loss.

| Initialization Method | Many | Med | Few | All |
|---|---|---|---|---|
| Xavier | 0.6793 | 0.4204 | 0.1823 | 0.3784 |
| Sparse Xavier | 0.7089 | 0.4977 | 0.2189 | 0.4237 |
| Gassian | 0.6167 | 0.3858 | 0.1572 | 0.3407 |
| Sparse Gassian | 0.7119 | 0.4969 | 0.2117 | 0.4209 |
| Uniform | 0.6119 | 0.3631 | 0.1638 | 0.3366 |
| Sparse Uniform | 0.7141 | 0.4927 | 0.2174 | 0.4231 |
| Orthogonal | 0.6844 | 0.4446 | 0.1868 | 0.3882 |
| Sparse Orthogonal | 0.7022 | 0.4846 | 0.2060 | 0.4124 |
| Kaiming Uniform | 0.6933 | 0.4392 | 0.1881 | 0.3898 |
| Sparse Kaiming Uniform | 0.7130 | 0.4946 | 0.2168 | 0.4230 |

Table 9: Sparsification with different initialization of frozen classifiers

necessitates substantial computational cost. However, owing to the advanced design of SSE-C, Fed-LoGe does not depend on the projection layer and dot-regression loss. We undertake experiments to investigate how the dot-regression loss and projection layer influence model performance based on ETF and FedLoGe, with the results documented in Tab.8. We ascertain that solely through the sparsity design of SSE-C, performance surpassing that of employing both the projection layer and dot-regression loss can be achieved.

## A.10 DIFFERENCE INITIALIZATION METHODS OF FIXED CLASSIFIER

To further explore the effectiveness of sparsification for a fixed classifier, we have selected various classifier initialization methods (Xavier, Gaussian, Uniform, Orthogonal, Kaiming Uniform), and subsequently applied sparsification to classifiers constructed with these initializations. The results are documented in Tab. 9. It is observable that irrespective of the initialization method employed, post-sparsification and subsequent training of the fixed classifier yield exceptionally favorable performance.

## A.11 DISCUSSION OF PRIVACY

In the entire training process of FedLoGe, no extra information compared with FedAvg is transmitted: 1. During the 'Representation Learning with SSE-C' phase, FedLoGe transmits local model weights as FedAvg necessitates. 2. In the 'Feature Realignment' phase, the local models are already downloaded for norm analysis, and there are no further information-sharing operations required.

As the norm analysis is performed on the client side or global side, there would still be a few privacy concerns. However, noteworthy that the potential privacy issue exists in the general FL frameworks rather than specific to our proposed Global and Local Adaptive Feature Realignment (GLA-FR). For instance, gradient inversion Huang et al. (2021) can pose a threat to almost all gradient transmission-based FL methods without any privacy-preservation techniques. As the privacy issue of the FL framework is beyond the scope of this work.

A.12   CONVERGENCE EXPERIMENTS

We have conducted experimental investigations into the convergence. Specifically, we explored the number of rounds required to reach an accuracy of 0.33 on CIFAR100 with an imbalance factor (IF) of 100 and $\alpha = 0.5$, as documented in the table below. It is evident that the convergence rate of FedLoGe is almost on par with FedAvg and surpasses the current state-of-the-art (SOTA) methods.

| Methods | Convergence Round |
|---|---|
| FedETF | 83 |
| FedROD | 68 |
| FedLoGe (Ours) | 62 |

Table 10: Comparison of Convergence Rounds for Different Methods.

A.13   COMPARISON OF EXISTING METHODS

We are the first to achieve joint global and local training in Federated long-tailed learning from the perspective of neural collapse. In the introduction, we analyzed the improvements of FedLoGe over some existing methods, and here we further provide the discussion:

First, most existing methods in personalized federated learning (Collins et al. (2021); Li et al. (2021a); T Dinh et al. (2020); Zhang et al. (2022b); Dai et al. (2022)) utilize the global model to regularize or construct personalized models but do not evaluate them in a generic setup, failing to yield a strong global model without long-tail bias. This makes them unsuitable as a strong incentive for attracting new clients. In contrast, our approach learns models that excel in both personalized (GA-FR) and generic (FA-LR) setups without compromising either.

Second, existing methods in federated long-tailed learning (Shang et al. (2022b); Yang et al. (2023a); Shang et al. (2022a); Wang et al. (2022); Qian et al. (2023)), focusing only on calibrating long-tail bias for a single global model, overlook the aspect of representation learning (backbone) under federated long-tailed settings and fail to provide personalized models. For instance, CreFF involves post-classifier retraining but does not advance in boosting the backbone and does not offer personalized models. In contrast, Our FedLoGe achieves strong representation learning with SSE-C to support both global model and local models.

Second, existing work involving fixed classifiers has not delved deeply into the analysis of the initialization of these fixed classifiers. FedBABU (Oh et al. (2022)), for instance, merely employs a randomly initialized fixed linear layer. For FedETF (Li et al. (2023)), the Vallina fixed ETF classifier has suffered from feature degeneration, characterized by significant fluctuations between features and the class mean, while FedLoGe alleviates this issue by filtering out noisy smaller mean features (Fig. 1), making features more concise and effective (Fig. 3 (a,b) and Fig. 6). Besides, FedLoGe is less computationally expensive and easier to adapt to existing algorithms. DR loss is necessary for FedETF, which means some modifications are required when integrating with existing algorithms. Besides, FedETF needs a projection layer for optimization. For ResNet18 with four blocks, a 512x512 projection layer would be approximately 1MB, almost doubling the parameter size of Block 1, which is 0.56MB.

A.14   IMPACT OF CLIENT NUMBERS AND PARTICIPATION RATES

To investigate the adaptability and efficiency of FedLoGe in varied federated learning environments, we designed experiments focusing on two main variables: the number of clients and their participation rates. The experiments spanned across CIFAR10/100-LT and ImageNet/iNat datasets, tailored to simulate environments ranging from full participation to more realistic, sporadic engagement.

For CIFAR10/100-LT, we utilized a setup involving 40 clients with 100% participation and 5 local epochs per training round. Conversely, ImageNet/iNat experiments were conducted with 20 clients at a 40% participation rate and 3 local epochs, introducing a scenario with reduced participation and increased data complexity.

Additionally, to explore the behavior of FedLoGe under more stringent conditions, we conducted experiments on CIFAR100-LT with an imbalance factor of 100 and a heterogeneity coefficient of 0.5, decreasing the participation rate to 30%. This setup aimed to test FedLoGe's resilience and personalized performance against data imbalance and client heterogeneity. Extending our exploration to a cross-device FL setting, the client count was increased to 100 with a 30% participation rate, challenging FedLoGe with a highly distributed environment and testing its scalability and performance.

The experimental outcomes, detailed in Table 11, highlight FedLoGe's superior performance across different settings, particularly showcasing its strengths in scenarios with varied client participation rates and numbers.

| Method | 40 clients (100% participate) | 40 clients (30%) | 100 clients (30%) |
|---|---|---|---|
| FedAvg | GM0.3818/PM0.6214 | 0.3746/0.6205 | 0.3718/0.6384 |
| FedRoD | 0.3919/0.6919 | 0.3758/0.7156 | 0.3727/0.7665 |
| FedETF | 0.3825/0.6421 | 0.4180/0.7083 | 0.3847/0.7822 |
| FedLoGe | **0.4233/0.7285** | **0.4206/0.7524** | **0.4159/0.8008** |

Table 11: Impact of client diversity and participation rates on FedLoGe performance compared to other FL algorithms.

These results confirm FedLoGe's robustness and adaptability, underlining its potential for broad applicability in federated learning scenarios characterized by varying degrees of client engagement and diversity.

### A.15 EFFECTIVENESS OF SSE-C

We opted for a learning rate of 0.0001 with 10,000 optimization steps on ImageNet-LT with 1000 classes. The order of magnitude for the norm variance is between $10^{-7}$ and $10^{-9}$. For the angles across each pair of classifier vectors, we visualized the cosine similarity in SSE-C in Fig. 7, indicating that our optimized SSE-C is almost identical to the dense ETF.

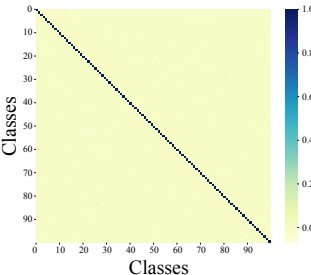

Figure 7: The heatmap of cosine similarity between classifier vectors in SSE-C.

