# OpenReview forum: "FedLoGe: Joint Local and Generic Federated Learning under Long-tailed Data"
_ICLR.cc/2024/Conference — ICLR 2024 poster_

### Official Review · Reviewer_YQxK · 2023-10-28

**Soundness:** 3 good
**Presentation:** 2 fair
**Contribution:** 2 fair
**Rating:** 5
**Confidence:** 3

**Summary:**

This paper aims to address the long-tailed data in FL. The authors propose a new method called FedLoGe to improve both local and global model performance of FL under this scenario. FedLoGe introduces a Static Sparse Equiangular Tight Frame Classifier (SSE-C) to enhance data representations. It also contains feature realignment at both global and local levels.

**Strengths:**

- The long-tailed problem in FL is an emerging problem that is worth studying.
- The paper is generally well structured.
- Figure 2 provides a good illustration of the proposed method.
- The proposed method seems to work and the experiments demonstrate the effectiveness of the proposed method.
- Evaluation are conducted on various backbones and different scales of datasets.

**Weaknesses:**

- The content of the paper is not very easy to follow. For example, the message from Figure 1 is a bit hard to interpret. In the first paragraph of the third page (where Figure 1 (a) is explained), it is not quite clear what large mean and small mean imply.
- Notations seem to be inconsistent, the notation for the number of clients is K in Section 3, but N in Section 4.
- Probably a typo: “…. personalized classifier by multiple the norm of … “ Should it be “by multiplying… “?
- The method of fixing the classifier and backbone alternatively for training is proposed in previous papers on personalized FL.
- Some settings are not clear: e.g., the number of clients participating in training each round and the number of local epochs.

**Questions:**

- How many clients participate in training each round?
- Does the paper focus on cross-silo FL, cross-device FL, or both?
- Why the proposed method can improve performance on FL with long-tailed problem?

---

> ### Author Response · Authors · 2023-11-17
> **Response for Reviewer YQxK[1/2]**
>
> # Response for R4
>
> Many thanks for your detailed and valuable comments. I have addressed each of your points in the following responses.
>
> Q1. What does Figure 1 signify, and what are the implications of large and small means?
>
> Thanks for the question and we are glad to clarify what Fig. 1 signifies: Compared to Fig. 1 (a), **two significant changes** are evident in Fig. 1 (b) after implementing FedLoGe: **1. Noisy features are masked.** The small mean features with bigger variances are located in the sparsely allocated area, marked by grey vertical shadows. 2. **The quality of dominant features is further enhanced**. This is evidenced by the decreased variance in features with larger means, indicating they have become more concise and effective.
>
> Implications of large and small means: Features with both **large and small means** are crucial for a network. However, training directly with a fixed ETF classifier can lead to poor quality of small mean features. This is reflected in the relative variance of smaller mean features within the same class being much higher compared to large mean features.
>
> Below is a detailed interpretation of Fig. 1:
>
> Neural collapse reveals a principle, namely feature collapse, which shows that in a properly trained network, features should effectively collapse into the class mean (**where all relative variances in Fig. 1 (a) are expected to be zero**). Yet, when trained with a vallina fixed ETF classifier, **features, particularly those with smaller means, do not collapse effectively, namely feature degeneration. FedLoGe is designed to mitigate feature degeneration by squeezing poor-quality and noisy features into these sparsely allocated positions.** Overall, as shown in Fig. 1, after training with $\psi_{SSE-C}$, FedLoGe inherently filters out noisy smaller mean features, making features more concise (Appendix A.4) and effective (Table 3 in the main paper), in which the variance of dominant features becomes smaller.
>
> We give the detailed computation process for the data in Fig. 1:
> Given that $n_c$ is the sample number of class $c$.  $\mu_c$ is the class mean of class $c$, and $h_{i,c}$ is the feature of class $c$ on sample $i$, the features $h_{i,c}$ will collapse to the within-class mean $\mu_c = \frac{1}{n_c} \sum_{i=1}^{n_c}h_{i,c}$. This indicates that the covariance $\frac{1}{n_c} \sum_{i=1}^{n_c}(h_{i,c}-\mu_c)(h_{i,c}-\mu_c)^T$ will converge to $0$. We investigate whether $h_{i,c} - \mu_c$ tends toward zero when training with a fixed ETF classifier, as shown in Fig. 1 (a).
>
> The steps to obtain the data in Fig. 1 are：
>
> 1. Select class $c$.
> 2. During global test, compute all sample features $h_{i,c}$, with dimension $D$, representing the number of features. Use $h_{i,c}[d]$ to denote the $d^{th}$ feature in $h_{i,c}$.
>
>  3.  Compute the mean and relative variance of $h_{c}$, sorting them in descending order of the mean, and accordingly adjust the variance positions.
>
> It becomes evident that not all features, $h_{i,c}[d] - \mu_c[d]$, converge to zero, and smaller $\mu_c[d]$ values exhibit greater variance.
>
> The process illustrated in **Fig.1 (b)** follows the same steps as **Fig.1 (a)**, but with grey vertical shadows for the sparsely positioned features. It can be observed that sparse ETF training effectively squeezes poor-quality and noisy features into these sparsely allocated positions.
>
> **All the calculation details have been updated in the revised appendix A.2 marked in red.**
>
> Q2. Notations and typos?
>
> Thank you for the detailed feedback! We have corrected all the inconsistent usage of K and N and revised “…. personalized classifier by multiple the norm of … “ to “by multiplying… “ in Section 3.3. and have uploaded the revised version and **marked the modifications in red color.**
>
> Q3. The method of fixing the classifier and backbone alternatively for training is proposed in previous papers on personalized FL.
>
> We need to clarify that the alternative training framework is a well-established training paradigm, inspired by decoupling learning [4]. **However, the main contribution of FedLoGe lies in being the first to achieve joint global and local training in Federated long-tailed learning, approached from the perspective of neural collapse**, utilizing the novel methods, $\psi_{SSE-C}$ and GLA-FR.
>
> We detail the differences between FedLoGe and traditional alternative training methods in personalized FL: Traditional alternative training for personalized FL **ignores representation learning for backbone**, making it **challenging to obtain global model and personalized models simultaneously**, as a strong backbone is foundational for both. Moreover, other personalized FL methods are **not specifically tailored for long-tail and heterogeneous data**.
>
> For example, FedRep [1] alternatively trains the classifier and backbone within a single local round. However, it does not specifically address imbalanced and long-tailed learning.

---

> ### Author Response · Authors · 2023-11-17
> **Response for Reviewer YQxK[2/2]**
>
> CreFF (Fed-LT) in [2] involves classifier retraining but does not make advancements in representation learning, nor does it provide personalized models. In contrast, FedLoGe is capable of **simultaneously training both generic and local models**. Notably, FedLoGe **integrates strong representation learning** [8] and **feature realignment to mitigate long-tailed and heterogeneous biases** through norm calibration [9, 10], all within the framework of neural collapse.
>
> **Please refer to the detailed discussion of the differences between our work and previous work in the responses to Reviewer tSL5 (W4), or in Section A.11 of our paper.**
>
> Q4. Some settings are not clear: e.g., the number of clients participating in training each round and the number of local epochs. How many clients participate in training each round? Does the paper focus on cross-silo FL, cross-device FL, or both?
>
> Thank you for detailed suggestions that really help in refining our work. FedLoGe covers diverse system scale in different dataset. For CIFAR10/100-LT datasets, we engaged 40 clients with full participation in each training round, setting the number of local epochs to 5. For ImageNet/iNat datasets, the experiments involved 20 clients, with a 40% participation rate in each training round, and the number of local epochs was set to 3.  **We have updated Section A.1, Detailed Setup, and marked the modifications in red for clarity.**
>
> To assess FedLoGe's resilience and effectiveness in diverse settings,  we conducted experiments on CIFAR100-LT with imbalance factor of 100 and heterogeneity coefficient of 0.5, while lowering the participation rate to 30%. The results, documented in the table below, shows that our approach remains superior, particularly in terms of personalized performance.
>
> To further explore the effectiveness of FedLoGe in a cross-device FL setting, we increased the number of clients to 100. The results demonstrate that FedLoGe significantly outperforms other algorithms.
>
> | Method | 40 clients(100% participate) | 40(30%) | 100(30%) |
> | --- | --- | --- | --- |
> | FedAvg | GM0.3818/PM0.6214 | 0.3746/0.6205 | 0.3718/0.6384 |
> | FedRoD | 0.3919/0.6919 | 0.3758/0.7156 | 0.3727/0.7665 |
> | FedETF | 0.3825/0.6421 | 0.4180/0.7083 | 0.3847/0.7822 |
> | FedLoGe | 0.4233/0.7285 | 0.4206/0.7524 | 0.4159/0.8008 |
>
> >Q5. Why the proposed method can improve performance on FL with long-tailed problem? （GLA-FR）
>
> Thanks for the question. Here's a detailed explanation of the mechanisms involved:
>
> **The classifier's class vector norm positively correlates with its predictive power and the proportion of that class's samples in the overall training set.**  The pattern has been noted in references [4, 5, 6]. Reference [7] offers an empirical explanation for this pattern in terms of decision boundaries. Additionally, within the neural collapse framework, references [9, 10] provide theoretical justification for the relationship between the norm and data distribution, based on the peer model.
>
> **The global model trained on long-tailed often exhibits biases**, favoring predictions for head classes. GA-FR addresses this by implementing self-normalization, dividing each class vector by its own norm to make them unit vectors in order to encourage the model to make more fair judgments.
>
> As seen in **Fig.2, the norm of $\psi$ becomes more balanced distributed after Global Adaptive Feature Realignment (GA-FR).** For experiments, we have reported improvements in Tables 1 and 2 and there are incredible performance enhancements for few and medium classes.
>
> It is important to note that Local Adaptive Feature Realignment (LA-FR) is designed to tailor the local model to fit local data, which is heterogeneous rather than typically long-tailed.
>
>
> More discussion on the motivation and effectiveness of using the integrated personalized classifier for each client as well as the feature alignment can be found in the response of Reviewer Nmh4, Q2, and Reviewer YHb3, Q3.
>
> [1] Exploiting Shared Representations for Personalzied Federated Learning. PMLR 2021.
>
> [2] Federated Learning on Heterogeneous and Long-Tailed Data via Classifier Re-Training with Federated Features. IJCAI 2022.
>
> [3] Fedic: Federated learning on non-iid and long-tailed data via calibrated distillation. IEEE ICME 2022.
>
> [4] Decoupling representation and classifier for long-tailed recognition. ICLR 2020.
>
> [5] Equalization loss v2: A new gradient balance approach for long-tailed object detection. CVPR 2021.
>
> [6] Over-coming classifier imbalance for long-tail object detection with balanced group softmax. CPVR 2020.
>
> [7] Adjusting decision boundary for class imbalanced learning. IEEE Access 2020.
>
> [8] Prevalence of neural collapse during the terminal phase of deep learning training. PNAS 2020.
>
> [9] Neural collapse in deep linear network: From balanced to imbalanced data. ICML 2023.
>
> [10] Imbalance trouble: Revisiting neural-collapse geometry. NIPS 2022.

---

> > ### Comment · Reviewer_YQxK · 2023-11-22
> >
> > Thank you for the rebuttal. Most of my concerns are addressed. However, I am still a bit concerned about Figure 1. The responses (also responses to other reviewers) help improve understanding, but it would be hard for a fresh reader to interpret the figure with only the description provided in the paper.
> >
> > After reviewing the revised manuscript again, it seems that not all citations are in the correct format. For example, "We examine the effectiveness of training with fixed classifiers in Fed-LT from the perspective of neural collapse (NC) Papyan et al. (2020)." Should the citation be like (Papyan et al., 2020)?

---

> ### Author Response · Authors · 2023-11-22
> **Further Response for Reviewer YQxK**
>
> Thank you for your detailed suggestions for revision. We have **redrawn Fig. 1** and added a detailed explanation of its calculation process in **Appendix A.2**. Additionally, we have **rewritten the caption** of Fig. 1 for clarity and included a **footnote on Page 2** to explain the implications of Fig. 1. These revisions have been **highlighted in red** and may help fresh readers to understand.
>
> Furthermore, we have meticulously reviewed all citations and corrected their formatting.
>
> Please refer to our revised version for all changes. We are grateful for your feedback and will continue to refine our work.

---

> ### Author Response · Authors · 2023-11-23
> **Kind Reminder: Review Deadline Approaching**
>
> Dear Reviewer YQxK:
>
> As the review deadline nears, with just six hours remaining, we wish to briefly highlight our **recent latest revision**.  In the latest update, we have focused on enhancing the clarity of Fig. 1 and have meticulously revised our citations to ensure full compliance.
>
> We hope these updates facilitate your re-evaluation. Your insights are invaluable to us, and we deeply appreciate your time and attention.
>
> Warm regards,
>
> Authors

---

> > ### Comment · Reviewer_YQxK · 2023-11-23
> >
> > Thank you for the response, and they resolve most of my concerns. I will decide on the final rating in the next phase after discussing it with other reviewers.

---

### Official Review · Reviewer_tSL5 · 2023-10-31

**Soundness:** 3 good
**Presentation:** 3 good
**Contribution:** 2 fair
**Rating:** 6
**Confidence:** 4

**Summary:**

This paper presented a model training framework that enhances the performance of both local and generic models in Fed-LT settings in the unified perspective of neural collapse. The proposed framework is comprised of SSE-C, a component developed inspired by the feature collapse phenomenon to enhance representation learning, and GLA-FR, which enables fast adaptive feature realignment for both global and local models. As a result, the proposed method attains significant performance gains over current methods in personalized and long-tail federated learning.

**Strengths:**

S1. This work focuses on addressing the data imbalance issue to simultaneously enhance the efficacy of the generic global model and its performance at the local level, which is an interesting topic.

S2. This paper is well-written and has a good presentation.

S3. The proposed SSE-C can address the problem of feature degeneration well, which is a promising finding.

**Weaknesses:**

W1. Convergence analysis is missing, which is very important for the optimization process of federated learning. Please analyze it from the experimental and theoretical point of view.

W2. Lack of discussion about privacy. Federated learning is proposed to protect the privacy of the client, but the method in this paper has the risk of gradient disclosure, please add the discussion of privacy in this work.

W3. The communication overhead of the model seems to be very large, which will restrict the practical application. Please increase the experiment and analysis of the communication overhead.

W4. The related work is simply a list of existing methods. Please add a discussion of the differences between this work and previous work to further clarify the contribution of this paper.

**Questions:**

Please see the Weaknesses.

---

> ### Author Response · Authors · 2023-11-17
> **Response for Reviewer tSL5 [1/2]**
>
> Thank you for providing suggestions that have helped in refining and improving our work from various perspectives.
>
> > W1. Convergence analysis is missing, which is very important for the optimization process of federated learning. Please analyze it from the experimental and theoretical point of view.
>
> We thank the reviewer for the thoughtful suggestion. We agree that the theoretical analysis would benefit the significance of the proposed method. However, given the limited time for rebuttal, we could not include a complete convergence analysis in this response. Drawing inspiration from neural collapse, a rich body of derivations about convergence proofs, based on the unconstrained features model and peel model, can be found in references [14, 15, 16, 17, 18, 19, 20, 21]. Based on this, we can extend these proofs to the federated framework and we shall provide the analysis in our revised supplementary material.
>
>
> While we are currently unable to provide formal proof of convergence, we have conducted experimental investigations into the convergence. Specifically, we explored the number of rounds required to reach an accuracy of 0.33 on CIFAR100 with an imbalance factor (IF) of 100 and alpha=0.5, as documented in the table below. It is evident that the convergence rate of FedLoGe is almost on par with FedAvg and surpasses the current state-of-the-art (SOTA) methods.
>
> | Methods | FedETF  | FedROD | FedLoGe (Ours) |
> | --- | --- | --- | --- |
> | Convergence Round | 83 | 68 | 62 |
>
> > W2. Lack of discussion about privacy. Federated learning is proposed to protect the privacy of the client, but the method in this paper has the risk of gradient disclosure, please add the discussion of privacy in this work.
>
> Thank you for highlighting the importance of discussing privacy aspects in our work. **In the entire training process of FedLoGe, no extra information compared with FedAvg is transmitted across clients and the server:** 1. During the 'Representation Learning with SSE-C' phase, FedLoGe transmits local model weights as FedAvg necessitates. 2. In the 'Feature Realignment' phase, the local models are already downloaded for norm analysis, and there are no further information-sharing operations required.
>
> As the norm analysis is performed on the client side or global side, there would still be a few privacy concerns. However, noteworthy that the potential privacy issue exists in the general FL frameworks rather than specific to our proposed Global and Local Adaptive Feature Realignment (GLA-FR). For instance, gradient inversion[1] can pose a threat to almost all gradient transmission-based FL methods without any privacy-preservation techniques.  Additional defense techniques such as Differential Privacy [22] and Homomorphic Encryption [23] could be considered to further enhance the performance of our work. As the privacy issue of FL framework is beyond the scope of this work, **we have included the discussion about privacy in the paper A.10, marked in red for clarity.**
>
> > W3. The communication overhead of the model seems to be very large, which will restrict the practical application. Please increase the experiment and analysis of the communication overhead.
>
> **In fact, FedLoGe ensures minimal communication overhead, as it does not require the transfer of any additional information beyond what is already exchanged in FedAvg.**
>
> For FedLoGe's computational cost, please refer to the experimental analysis presented in **Appendix A.3**. During the backbone training phase, using frozen $\psi_{SSE-C}$ means there’s no need to compute gradients in the fixed dense linear layer. Furthermore, feature realignment is a one-shot operation that requires only basic arithmetic computations. We conducted feature realignment (GLA-FR) for 40 clients and one global model on an NVIDIA GeForce RTX 3090, which took only 1.876e-2 seconds. Additionally, we assessed the computational expense of a single round of FedLoGe training on the PyTorch platform. The results presented in the table below show that FedLoGe is a lightweight algorithm like FedAvg and FedBABU.
>
> | Method | FedAvg | FedProx | FedRep | FedBABU | FedLoGe |
> | --- | --- | --- | --- | --- | --- |
> | Time Cost/Round | 3min11s | 5min24s | 4min07s | 3min14s | 3min19s |
>
>
> > W4. The related work is simply a list of existing methods. Please add a discussion of the differences between this work and previous work to further clarify the contribution of this paper.
>
> Thank you for pointing out the need for a more detailed discussion of related work comparisons. Your feedback will help us clarify the distinct contributions of our paper and **we have updated it in A.11** in red in the paper. To the best of our knowledge, w**e are the first to achieve joint global and local training in Federated long-tailed learning from the perspective of neural collapse.** In the introduction, we demonstrated the improvements of FedLoGe over some existing methods, and here we further provide the discussion:

---

> ### Author Response · Authors · 2023-11-17
> **Response for Reviewer tSL5 [2/2]**
>
> 1. Most existing methods in personalized federated learning [2, 3, 4, 5, 6] utilize the global model to regularize or construct personalized models but **do not evaluate them in a generic setup**, failing to yield a strong global model without long-tail bias. This makes them unsuitable as a strong incentive for attracting new clients. In contrast, our approach learns models that **excel in both personalized (LA-FR) and generic (GA-FR) setups** without compromising either.
> 2. Existing methods in federated long-tailed learning [7, 8, 9, 10, 11], focusing only on calibrating long-tail bias for a single global model, **overlook the aspect of representation learning (backbone) under federated long-tailed settings** and fail to provide personalized models. For instance, CreFF involves post-classifier retraining but does not advance in boosting the backbone and does not offer personalized models.  In contrast, Our FedLoGe **achieves strong representation learning with SSE-C** to support both global model and local models.
> 3. Existing work involving fixed classifiers has not delved deeply into the analysis of the initialization of these fixed classifiers. FedBABU[12], for instance, merely employs a randomly initialized fixed linear layer. For FedETF[13], the vallina fixed ETF classifier has **suffered from feature degeneration**, characterized by significant fluctuations between features and the class mean, while **FedLoGo alleviates this issue** by filtering out noisy smaller mean features (Fig. 1), making features more concise and effective (Fig. 3 (b, c) and Fig. 6). Besides, **FedLoGe is less computationally expensive and easier to adapt to existing algorithms.** DR loss is necessary for FedETF, which means some modifications are required when integrating with existing algorithms. Besides, FedETF needs a projection layer for optimization. For ResNet18 with four blocks, a 512x512 projection layer would be approximately 1MB, almost doubling the parameter size of Block 1, which is 0.56MB.
>
> [1] Evaluating gradient inversion attacks and defenses in federated learning. NIPS 2021.
>
> [2] Exploiting shared representations for personalzied federated Learning. PMLR 2021.
>
> [3] Ditto: Fair and robust federated learning through personalization. ICML 2021.
>
> [4] Personalized federated learning with moreau envelopes. NIPS 2020.
>
> [5] Personalized federated learning via variational bayesian inference. ICML 2022.
>
> [6] Dispfl: Towards communication-efficient personalized federated learning via decentralized sparse training. PMLR 2022.
>
> [7] Federated learning on heterogeneous and long-tailed data via classifier re-training with federated features. IJCAI 2022.
>
> [8] Integrating local real data with global gradient prototypes for classifier re-balancing in federated long-tailed learning. Arvix 2023.
>
> [9] Fedic: Federated learning on noniid and long-tailed data via calibrated distillation. IEEE ICME 2022.
>
> [10] Logit calibration for non-iid and long-tailed data in federated learning. IEEE ISPA/BDCloud/SocialCom/SustainCom 2022.
>
> [11] Long-tailed federated learning via aggregated meta mapping. IEEE ICIP.
>
> [12] FedBABU: Toward Enhanced Representation for Federated Image Classification ICLR 2021.
>
> [13] No fear of classifier biases: neural collapse inspired federated learning with synthetic and fixed classifier. CVPR 2023.
>
> [14] Prevalence of neural collapse during the terminal phase of deep learning training. PNAS 2020.
>
> [15] Neural collapse in deep linear network: From balanced to imbalanced data. ICML 2023.
>
> [16] Neural Collapse: A Review on Modelling Principles and Generalization. TMLR 2023.
>
> [17] Neural collapse with unconstrained features. Arvix 2020.
>
> [18] Neural collapse with normalized features: A geometric analysis over the Riemannian Manifold. NIPS 2022.
>
> [19] Exploring deep neural networks via layer-peeled model: Minority collapse in imbalanced training. PNAS 2021.
>
> [20] Fix your features: Stationary and maximally discriminative embeddings using regular polytope. Arvix 2019.
>
> [21] A geometric analysis of neural collapse with unconstrained features. NIPS 2021.
>
> [22] On privacy and personalization in cross-silo federated learning. NIPS 2022.
>
> [23] Privacy preserving machine learning with homomorphic encryption and federated learning. Future Internet 2021.

---

> > ### Comment · Reviewer_tSL5 · 2023-11-22
> >
> > Thank you for the rebuttal. Most of my concerns are addressed. Although my question about convergence analysis (W1) was not well answered, I  would raise my score to 6. Please incorporate the follow-up update into your paper.

---

> > > ### Author Response · Authors · 2023-11-22
> > > **Further Response for Reviewer tSL5**
> > >
> > > Thank you for your feedback. We will incorporate discussions on convergence, privacy, communication overhead, and comparative analysis in our updated paper. Thanks again for your support in improving our work.

---

### Official Review · Reviewer_YHb3 · 2023-11-01

**Soundness:** 2 fair
**Presentation:** 1 poor
**Contribution:** 3 good
**Rating:** 8
**Confidence:** 4

**Summary:**

The paper considers federated learning with long-tailed global distribution and non-iid client distributions. the authors propose a Federated Local and Generic Model Training (FedLoGe) to train a model that can perform well both on long-tailed global distribution and local client distributions. First, the proposed Static Sparse Equiangular Tight Frame Classifier fosters the acquisition of potent data representations. Second, the Global and Local Adaptive Feature Realignment (GLA-FR) is used to align global features with client preferences.

**Strengths:**

* (originality) The authors proposed a novel method by neural collapse and realignment of local and global classifiers. Though neural collapse has been used for handling data heterogeneity, the realignment is novel to address the local-generic problem.
* (significance) The proposed method can significantly outperform baselines in the concerned settings.
* (clarity) The figure 2 is appreciated for clarifying the method.
* (quality) The experiments are well designed for the problem setting. Multiple baselines are compared to demonstrate the effectiveness of the proposed method both in boosting local and generic performance.

**Weaknesses:**

* (quality) The generic-local setting looks self-contradictive to me. When the personalized federated learning targets training models personalized for each client, is it necessary to consider a model adapted for global or generic distribution? In personalized federated learning, the assumption is that each client has its own data distribution and it aims to learn a model adapted for the specific distribution. If the global or generic distribution is considered, which client will be expected to use the model?
* (significance) Because of my concerns about the problem setting, I am afraid that the proposed method may have a limited impact on a few specific problems.
* (clarity) The technical motivation is not clear. The neural collapse has been used for handling data heterogeneity and classifiers should be fixed by the neural collapse theorem. It is unclear why the realignment is still needed. Why the fixed classifier cannot be used for all clients?
* (clarity) The paper is hard to follow. Specifically, the motivation experiments in the introduction are hard to understand. Why the mean and variance of class means are evaluated in Figure 1? I cannot follow the logic in the discussion quoted below.
> However, preliminary experiments benchmarking ETF with CIFAR-100 in Fed-LT suggest that only a few features have relatively large means, while most of the small-mean features are contaminated by severe noise, as shown in Fig. 1(a). Such observations are inconsistent with the feature collapse property, and we coin it as feature degeneration.

  How do the mean and variance of features relate to the feature collapse?

**Questions:**

* Please provide clear motivation for the generic-local setting. When the personalized federated learning targets training models personalized for each client, is it necessary to consider a model adapted for global or generic distribution? In personalized federated learning, the assumption is that each client has its own data distribution and it aims to learn a model adapted for the specific distribution. If the global or generic distribution is considered, which client will be expected to use the model?
* How do the mean and variance of features relate to the feature collapse in the quoted content?
> However, preliminary experiments benchmarking ETF with CIFAR-100 in Fed-LT suggest that only a few features have relatively large means, while most of the small-mean features are contaminated by severe noise, as shown in Fig. 1(a). Such observations are inconsistent with the feature collapse property, and we coin it as feature degeneration.

=== after rebuttal ===
The authors' responses have address my concerns.

---

> ### Author Response · Authors · 2023-11-17
> **Response for Reviewer YHb3 [1/2]:**
>
> Thank you for your thoughtful review. I've carefully considered your points and respond to each as follows.
>
> > Q1. When the personalized federated learning targets training models personalized for each client, is it necessary to train a global model? Which client will be expected to use the model? Please provide clear motivation for the generic-local setting.
>
> Thank you for raising this important question about the necessity and role of a global model in personalized federated learning. We shall outline our motivation for the generic-local setting as follows:
>
> 1. **The global model plays a crucial role as it forms a synergistic and progressively enhancing relationship with the local models, leading to mutual improvements and advancements**. The global model becomes more robust and versatile by aggregating the learning outcomes of various local models, while the local models gain a better starting point from the updated global model, enabling them to learn the unique characteristics of their local data more effectively.
> 2.  **A high-quality global model acts as a strong incentive for attracting new clients to join the federated training**. When new clients join the training, they can directly access the quick benefits of a good generic model and continuously share local knowledge in the following training. Otherwise, training from scratch could be a cost-intensive option. Without well trained global model, existing local clients are unwilling to share their own private personalized models with the new clients who make no contributions to the training process.
> 3. **A high-quality global model supports faster local model adaptation**.  Taking into account scenarios where users opt to **purchase the global model for local adaptation**, rather than joining the federated training process. We conduct an experiment to investigate the local adaptation efficiency with a high-quality global model. Assuming 20 clients take a trained global model for local fast adaptation, we have documented the average accuracy of local models across various local adaptation epochs **with and without Global Adaptive Feature Realignment (GA-FR)**, as shown in the table below. It shows that employing GA-FR can achieve better performance in the same training epochs.
> 4. **Other personalized federated learning papers have also considered the generic-local setting [1, 2]**, though they do not specifically designed for long-tailed and imbalanced data.
>
> |  | epoch 15 | epoch 20 | epoch 25 |
> | --- | --- | --- | --- |
> | Vallina Global Model | 0.6117 | 0.6331 | 0.6441 |
> | Global Model with GA-FR | 0.6544 | 0.6606 | 0.6642 |
>
>
> > Q2. Given my concerns about the problem setting, could it be that the proposed method will have limited impact, affecting only a few specific problems?
>
> Thank you for your concern regarding the potential scope of impact of our proposed method. **Data privacy and long-tailed distribution are more common than exceptional in many real-world scenarios, and FedLoGe is applicable in these situations.** As more clients join the training process, FedLoGe not only offers excellent personalized solutions but also provides a superior global model that attracts even more clients, thereby creating a positive feedback loop.
>
> For instance, patients’ diagnosis varies substantially across medical centers but collaboratively form long-tailed distributions[3, 4]. FedLoGe can provide personalized models for specific local diagnostics, while also offering a high-quality global model that incentivizes other medical centers to participate and contribute to the training process.
>
>
> > Q3. Why the realignment is still needed. Why the fixed classifier cannot be used for all clients?
>
> **The realignment is still needed because a single fixed classifier is not suitable for all clients with diverse local data**. The fixed classifier is employed to train a robust backbone, while realignment further customizes the classifier, tailoring the model for specific local data. As illustrated in the table below, we experimented with various classifiers and the results clearly show that Local Adaptive Feature Realignment (LA-FR) is most effective.
>
> In the table, $\psi$ denotes the global classifier and $\phi_k$ denotes the classifier of the $k^{th}$ client. We derive the final prediction head for the $k^{th}$ client $\phi_k^{\prime}$ combine with $\phi_k$  and $\psi$ with $\phi'_{k,c}=\psi_c$
>
> $*\left\|\phi_{k,c}\right\|_2$.
>
> | $\phi_k$ | $\psi$ | $\psi_{SSE-C}$ | $\phi_k^{\prime}$ |
> | --- | --- | --- | --- |
> | 0.6986 | 0.7128 | 0.7146 | 0.7341 |
>
> Furthermore, the table also indicates that only relying on a personalized classifier does not yield optimal performance either. The local head, lacking global knowledge, tends to overfit to local data. In contrast, the combination of global and local heads, as implemented with our LA-FR, can leverage both the extensive global knowledge and the specific data statistics of the local context.

---

> ### Author Response · Authors · 2023-11-17
> **Response for Reviewer YHb3 [2/2]**
>
> More discussion on the motivation for using the integrated personalized classifier for each client can be found in the response of Reviewer Nmh4, Q2.
>
>
> > Q4. Why the mean and variance of class means are evaluated in Figure 1? How do the mean and variance of features relate to the feature collapse in the quoted content?
>
> Thanks for the questions. The objective of analyzing variance and means of features is to examine whether the properties revealed by neural collapse theory are maintained when using a fixed ETF classifier.
>
> **Relations bewteen feature collapse with mean and variance of features:** Neural collapse reveals a principle, namely feature collapse, which shows that in a properly trained network, features should effectively collapse into the class mean (**where all relative variances in Fig. 1 (a) are expected to be zero)**. Yet, when trained with a vallina fixed ETF classifier, **features, particularly those with smaller means, do not collapse effectively, namely feature degeneration. FedLoGe is designed to mitigate feature degeneration by squeezing poor-quality and noisy features into these sparsely allocated positions.** Overall, as shown in Fig. 1, after training with $\psi_{SSE-C}$, FedLoGe inherently filters out noisy smaller mean features, making features more concise (Appendix A.4) and effective (Table 3 in the main paper), in which the variance of dominant features becomes smaller.
>
> **Main changes between Fig. 1 (a) and (b):** Compared to Fig. 1 (a), **two significant changes** are evident in Fig. 1 (b) after implementing FedLoGe: **1. Noisy features are masked.** The small mean features with bigger variances are located in the sparsely allocated area, marked by grey vertical shadows. 2. **The quality of dominant features is further enhanced**. This is evidenced by the decreased variance in features with larger means, indicating they have become more concise and effective.
>
> We give the detailed computation process for the data in Fig. 1:
> Given that $n_c$ is the sample number of class $c$.  $\mu_c$ is the class mean of class $c$, and $h_{i,c}$ is the feature of class $c$ on sample $i$, the features $h_{i,c}$ will collapse to the within-class mean $\mu_c = \frac{1}{n_c} \sum_{i=1}^{n_c}h_{i,c}$. This indicates that the covariance $\frac{1}{n_c} \sum_{i=1}^{n_c}(h_{i,c}-\mu_c)(h_{i,c}-\mu_c)^T$ will converge to $0$. We investigate whether $h_{i,c} - \mu_c$ tends toward zero when training with a fixed ETF classifier, as shown in Fig. 1 (a).
>
> The steps to obtain the data in Fig. 1 are：
>
> 1. Select class $c$.
> 2. During global test, compute all sample features $h_{i,c}$, with dimension $D$, representing the number of features. Use $h_{i,c}[d]$ to denote the $d^{th}$ feature in $h_{i,c}$.
>
>  3.  Compute the mean and relative variance of $h_{c}$, sorting them in descending order of the mean, and accordingly adjust the variance positions.
>
> It becomes evident that not all features, $h_{i,c}[d] - \mu_c[d]$, converge to zero, and smaller $\mu_c[d]$ values exhibit greater variance.
>
> The process illustrated in **Fig.1 (b)** follows the same steps as **Fig.1 (a)**, but with grey vertical shadows for the sparsely positioned features. It can be observed that sparse ETF training effectively squeezes poor-quality and noisy features into these sparsely allocated positions.
>
> **All the calculation details have been updated in the revised appendix A.2 marked in red for clarity.**
>
> [1] On Bridging Generic and Personalized Federated Learning for Image Classification. ICLR 2022.
>
> [2] Spectral Co-Distillation for Personalized Federated Learning. NeurIPS 2023.
>
> [3] lluminating the dark spaces of healthcare with ambient intelligence. Nature 2020.
>
> [4] Gender imbalance in medical imaging datasets produces biased classifiers for computer-aided diagnosis. PNAS 2020.

---

> ### Comment · Reviewer_YHb3 · 2023-11-22
>
> Thanks for the rebuttal.
>
> Q1. Motivation for the generic-local setting
>
> I agree that a high-quality global model could speed up convergence and incentivize new users. But the problem is **how to define the quality of global models**. Currently, the quality is defined as the performance of the global model on global data distribution. However, I doubt that good performance on long-tailed global distribution could be preferred for new users or any users on the tail. Intuitively, tail (new) users prefer a global model that can work well (out-of-box) on their own distribution. Obviously, performance on global distribution can not provide such attraction.
>
> In contrast, good local model performance for all users with a low variance is a better quality measure. New users would love to see all users gain an advantage, and so will themselves.
>
> Q2: Limited impact due to my concern in Q1.
>
> As I am still concerned about the motivation for a generic-local setting, I keep my concern for the impact.
>
> Q3: Why the realignment is still needed.
>
> The result is convincing for using a mixture of local and generic classifiers.
>
> Plus, I am slightly confused about why the generic-local intuition conflicts with the Neural Collapse (NC)? According to NC, shouldn't we use a fixed classifier for all clients? As the authors emphasize a lot on NC as an intuition, the conflict should be explained more intuitively.
>
> Q4. Why the mean and variance of class means are evaluated in Figure 1?
>
> Thanks for the updates, which is clearer now.

---

> > ### Author Response · Authors · 2023-11-22
> > **Further Response for Reviewer YHb3**
> >
> > Thank you for your in-depth discussion and comments on our work.
> >
> > > Q1 & Q2, Motivation for the generic-local setting.
> > >
> >
> > Thank you for your question regarding the quality of a global model.
> >
> > **In the context of long-tailed learning, a high-quality model typically refers to one that, despite being trained on long-tailed data, exhibits strong performance on a balanced test set**, as supported by references [1,2,3,4,5,6]. The objective of long-tailed learning is to eliminate long-tail biases and obtain a balanced model for future use. **Our goal of global realignment aligns with the objective of long-tailed learning**; we aim to train a global model that performs well on a balanced dataset, matching the objective of all existing long-tail learning works.
> >
> > Returning to the federated learning scenario, **while the advent of numerous personalized models has enhanced applicability, it does not diminish the significance of training a globally balanced model.** Methods targeting imbalance[7, 8, 9, 10] in federated learning are still crucial in achieving a balanced model with good generalization capability.
> >
> > Back to our method, concerning tail (new) users, they do prefer a global model that works exceptionally well on their own distribution right out of the box, but, currently, there is no existing model available that is out-of-box ready for new users. Our balanced global model can **serve as a more equitable starting point that enables fast local adaptation for all classes**. Intuitively, when we cannot presume the data distribution of new clients, **providing a model with the most generalizable and equitable performance – a balanced model – becomes the choice with safety and greater overall expected benefits**. To illustrate the case, consider a global model with a severe long-tailed bias provided to users on the tail users. Given the limited information about tailed classes in the global classifier without global realignment, it would be challenging, if not impossible, for them to learn effectively, regardless of the efforts made.
> >
> > > Q3: Why the realignment is still needed?
> > >
> >
> > Thank you for your query. The concept of generic-local intuition is indeed in harmony with Neural Collapse (NC) theory. Our approach, Global Local Adaptive Feature Realignment **(GLA-FR), also draws inspiration from imbalanced NC [11, 12]** (as outlined in Introduction, C2, in the main paper), which prompted us to develop classifiers for different clients. Specifically, NC theory suggests the need for feature realignment: it indicates that categories with fewer samples in the training set correspond to class vectors with smaller norms, leading to diminished predictive strength. By adjusting the norm of local class vectors, we align them with the local data distribution. **Our** **fixed global classifier $\psi_{SSE-C}$ and the GLA-FR are seamlessly integrated under the cohesive framework of Neural Collapse (NC)** — one is designed to enhance the backbone, while the other calibrate the classifier to further boost the performance.
> >
> > Additionally, when the NC concept, established within centralized learning, is applied to federated contexts, **a more practical and important challenge arises: a single, fixed classifier falls short for clients with heterogeneous local data.** This challenge calls for a redesign through feature realignment. However, during realignment, if the norms of the ETF classifier are uniform, they do not reflect local data specifics. Consequently, we have introduced additional classifiers to aid in the feature realignment.
> >
> > > Q4. Why the mean and variance of class means are evaluated in Figure 1?
> > >
> >
> > Thank you for your assistance in improving our work. In addition to the appendix, we will also include more explanations in the revised version of the main paper.
> >
> > [1] Deep long-tailed learning: A survey, TPAMI 2023
> >
> > [2] Long-tailed classification by keeping the good and removing the bad momentum causal effect NIPS2020
> >
> > [3] Targeted supervised contrastive learning for long-tailed recognition. CVPR 2022.
> >
> > [4] Federated Learning on Heterogeneous and Long-Tailed Data via Classifier Re-Training with Federated Features. IJCAI 2022.
> >
> > [5] Fedic: Federated learning on non-iid and long-tailed data via calibrated distillation. IEEE ICME 2022.
> >
> > [6] Fed-GraB: Federated Long-tailed Learning with Self-Adjusting Gradient Balancer. NIPS 2023.
> >
> > [7] Federated learning on heterogeneous and long-tailed data via classifier re-training with federated features. IJCAI 2022.
> >
> > [8] Fedic: Federated learning on noniid and long-tailed data via calibrated distillation. IEEE ICME 2022.
> >
> > [9] Addressing Class Imbalance in Federated Learning, AAAI 2021.
> >
> > [10] Fed-GraB: Federated Long-tailed Learning with Self-Adjusting Gradient Balancer, NIPS 2023.
> >
> > [11] Neural collapse in deep linear network: From balanced to imbalanced data. ICML 2023.
> >
> > [12] Imbalance trouble: Revisiting neural-collapse geometry. NIPS 2022.

---

> > > ### Comment · Reviewer_YHb3 · 2023-11-22
> > >
> > > Thanks for the responses. All of my concerns have been addressed.
> > >
> > > Please incorporate the clarifications into your paper. Could you highlight the revised content to reflect the clarification on our discussed problems?

---

> ### Author Response · Authors · 2023-11-22
> **Further Response for Reviewer YHb3**
>
> Thank you for your feedback. We will incorporate the requested clarifications into our paper. Thank you again for your valuable comments.

---

### Official Review · Reviewer_Nmh4 · 2023-11-08

**Soundness:** 3 good
**Presentation:** 2 fair
**Contribution:** 2 fair
**Rating:** 5
**Confidence:** 3

**Summary:**

This paper addresses Federated Learning with global long-tailedness. The proposed method is inspired by the neural collapse idea and contains three steps: 1) global representation learning with SSE-C classification head, and 2) & 3) global and local feature realignment.  In step 1, a classifier head is learned with sparsity while maintaining the Equiangular Tight Frame (ETF) properties, which is then broadcast to clients for learning backbone $\theta$ and personalized classifier head $\psi$. In step 2 & 3, the backbone $\theta$ and global classifier $\psi$ and local classifier $\phi$ is respectively updated and realigned. Empirical results on a few benchmarks show the effectiveness of the proposed approach over prior work on data heterogeneous FL or Long-tailed FL.

**Strengths:**

\+ This paper tackles a crucial and intriguing problem in federated learning with global long-tailedness.

\+ The key idea of using ETF to build a classifier as regularized consensus is well-motivated.

\+ Experiments and ablation studies are well-designed and comprehensively conducted.

**Weaknesses:**

\- Too many technical details are missing. For example, it is vague to me why performing personal classifier realignment in Eq (7) helps in tackling local data heterogeneity.

\- Authors need to provide detailed motivation of each step in the proposed algorithm, such as the motivation of keeping two classifier heads for clients: $\phi_k$ and $\psi_k$, and realignment of the global classifier $\psi$ in Eq 6.

\- Writing: Certain citation formats need to be fixed; Figure 1 is difficult to comprehend.

\- Missing discussion and comparison with related work on long-tailed FL. Authors should specify the technical contributions of the proposed method over other prior work, especially FedETF, which shows much resemblance to the proposed work.

**Questions:**

How to optimize Eq (4) that contains a maximization over $j$?

What is the purpose of introducing sparsity to the SSE-C classifier, besides its empirical effectiveness?

Why do we need two classifier heads  $\phi_k$ and $\psi_k$ for each of the clients $k$ instead of just one?

---

> ### Author Response · Authors · 2023-11-17
> **Response for Reviewer Nmh4 [1/3]:**
>
> Thank you for your insightful feedback. I am pleased to address each of your points in the following responses.
>
> > Q1. What is the detailed motivation of each step in the proposed algorithm, such as 1. why does performing personal classifier realignment in Eq. 7 help in tackling local data heterogeneity? 2. what’s the motivation of keeping two classifier heads for clients and the realignment of the global classifier in Eq. 6?
>
> Thank you for your insightful questions regarding the detailed motivations behind the steps in our proposed algorithm. I am pleased to explain the motivations point by point. FedLoGe achieves joint training of generic and local models by decoupling representation learning (backbone) and classifier learning [1, 2] in sequential order:
>
> Step 1: During the backbone training phase, we employ the fixed SSE-C classifier to address the divergence caused by data heterogeneity. A well-trained shared backbone is essential, as it lays the foundation for enhancing the potential of both the global model and local models.
>
> Step 2: In the feature realignment stage, Global and Local Adaptive Feature Realignment (GLA-FR) focuses on aligning the model's capabilities with the corresponding data distributions. Building on a robust backbone, we further customize the classifiers to fit corresponding test data. The global model is designed for fairly treating all classes (GA-LR), **highlighted in line 8 of the Revised Algorithm**. Meanwhile, the personalized models are aligned with the local statistics **(LA-FR, Eq. 7)**, seen in **Revised Algorithm, line 11.**
>
> Specifically, we explain the motivation and objective of the feature alignment of FedLoGe. **The objective of feature realignment (FR) is to address challenges brought by the heterogeneous and imbalanced data.** In the context of long-tailed learning, it is commonly acknowledged that the classifier's class vector norm positively correlates with its predictive power and the proportion of that class's samples in the overall training set[2, 3, 4, 5, 6, 7]. In Fed-LT, the long-tailed data would lead to a highly imbalanced classifier throughout the training process. Therefore, we use FR to re-balance the norm of the classifier for a balanced representation learning with heterogeneous global/local data.
>
> GA-FR is effective because it aligns the long-tailed norm to the balanced distribution caused by a long-tailed training set, thus balancing the predictive abilities of all classifiers. Due to the inconsistency of local and global data distribution, GA-FR fails to fit the local statistic. **We design LA-FR, which adapts the global classifier to fit the local statistics by local classifier norms.** We have provided a detailed explanation of keeping two classifier heads in response to Question 2.
>
> > Q2. Why do we need two classifier heads?  Global $\psi$ and local $\phi_k$ for each of the clients $k$ instead of just one ($\psi$ or $\phi_k$)?
>
> Notations:
> $\psi$ denotes global classifier and $\phi_k$ denotes the classifier of the $k^{th}$ client. We derive the final prediction head for the $k^{th}$ client $\phi_k^{\prime}$ combine with $\phi_k$ and $\psi$ with $\phi'_{k,c}=\psi_c*$
>
> $\left\|\phi_{k,c}\right\|_2$.
>
> Thanks for your questions about local classifier design. We need to clarify that the reason why we do not use global $\psi$ for each client is due to a single classifier is not suitable for clients with diverse local data. Besides, a single local head, due to its absence of global knowledge, is prone to overfitting to local data.
> **The integration/combination of the global and local heads leverages both the comprehensive knowledge from the global perspective and the data statistics specific to the local context.**
>
> The collaboration of global $\psi$ and local $\phi_k$ could be regarded as knowledge transfer: **transferring** global generalized knowledge with the norm of local $\phi_k$ **to** the new integrated local personalized head (The classifier's class vector norm positively correlates with its predictive power and the proportion of that class's samples in the overall training set. See Reviewer YQxK, Question 5 for details.).
>
> The visualization in Figure 2, particularly the sections illustrating 'transfer' and 'decline', highlights that local models prioritize classifiers for classes more common in their local data, thereby improving accuracy for these relevant classes.
>
> An extreme example is that for a class $c$ not present in a local model, it is reasonable to completely obliterate the predictive capability of classifier $c$. This corresponds to the combination process $\mathbf{\phi}^{\prime}_{k,c}$
>
> $=\mathbf{\psi}_{c} $
>
> $* \left\| \mathbf{\phi}_{k,c} \right\|_2 $
>
> $= \mathbf{\psi}_{c} * 0 = 0$.
>
> We further experimented to explore the performance of different schemes for client $k$ as illustrated in the following table. It show that scheme 3, $\phi'_{k,c}=$
>
> $\psi_c*\left\|\phi_{k,c}\right\|_2$, yields the most beneficial result.

---

> ### Author Response · Authors · 2023-11-17
> **Response for Reviewer Nmh4 [2/3]:**
>
> | 1. $\psi$ | 2. $\phi_k$ | 3. $\phi'_{k,c}$ | 4. $\phi'_{k,c} + \phi_k$ | 5. $\psi_{SSE-C}$ |
> | --------- | ----------- | ---------------- | ------------------------- | ----------------- |
> | 0.7128    | 0.6986      | 0.7341           | 0.7321                    | 0.7146            |
>
> > Q3. Fig. 1 is difficult to comprehend.
>
> Thanks for the question and we are glad to clarify what Fig. 1 implies. Compared to Fig. 1 (a), **two significant changes** are evident in Fig. 1 (b) after implementing FedLoGe: **1. Noisy features are masked.** The small mean features with bigger variances are located in the sparsely allocated area, marked by grey vertical shadows. 2. **The quality of dominant features is further enhanced**. This is evidenced by the decreased variance in features with larger means, indicating they have become more concise and effective.
>
> Below is a detailed interpretation of Fig. 1:
>
> Neural Collapse (NC) theory highlights a key characteristic of successful neural network training, which is the convergence of all sample features (outputs of the backbone) within the same category to the class mean of that category.
> **Yet, when trained with a vallina fixed ETF classifier (Fig. 1 (a)), features with smaller means do not collapse effectively, namely feature degeneration. FedLoGe(Fig. 1 (b)) is designed to mitigate feature degeneration by squeezing poor-quality and noisy features into these sparsely allocated positions.**
>
> Overall, as shown in Fig. 1, after training with $\psi_{SSE-C}$, FedLoGe inherently filters out noisy smaller mean features, making features more concise (Appendix A.4) and effective (Table 3 in the main paper), in which the variance of dominant features becomes smaller.
>
> We give the detailed computation process for the data in Fig. 1:
> Given that $n_c$ is the sample number of class $c$. $\mu_c$ is the class mean of class $c$, and $h_{i,c}$ is the feature of class $c$ on sample $i$, the features $h_{i,c}$ will collapse to the within-class mean $\mu_c = \frac{1}{n_c} \sum_{i=1}^{n_c}h_{i,c}$. This indicates that the covariance $\frac{1}{n_c} \sum_{i=1}^{n_c}(h_{i,c}-\mu_c)(h_{i,c}-\mu_c)^T$ will converge to $0$. We investigate whether $h_{i,c} - \mu_c$ tends toward zero when training with a fixed ETF classifier, as shown in Fig. 1 (a).
>
> The steps to obtain the data in Fig. 1 are：
>
> 1. Select class $c$.
> 2. During global test, compute all sample features $h_{i,c}$, with dimension $D$, representing the number of features. Use $h_{i,c}[d]$ to denote the $d^{th}$ feature in $h_{i,c}$.
>
> 3. Compute the mean and relative variance of $h_{c}$, sorting them in descending order of the mean, and accordingly adjust the variance positions.
>
> It becomes evident that not all features, $h_{i,c}[d] - \mu_c[d]$, converge to zero, and smaller $\mu_c[d]$ values exhibit greater variance.
>
> The process illustrated in **Fig.1 (b)** follows the same steps as **Fig.1 (a)**, but with grey vertical shadows for the sparsely positioned features. It can be observed that sparse ETF training effectively squeezes poor-quality and noisy features into these sparsely allocated positions.
>
> **All the calculation details have been updated in the revised appendix A.2 marked in red.**
>
> > Q4. Comparison with related work on long-tailed FL, specify the technical contributions of the proposed method over other prior work, especially FedETF, which shows much resemblance to the proposed work.
>
> Other long-tailed FL approaches [8, 9, 10, 11] focus on a single global model, overlooking personalized client needs. In contrast, FedLoGe is capable of simultaneously training both generic and local models. Notably, FedLoGe integrates representation learning and feature realignment under the unified framework of neural collapse.
> Below are the detailed differences between FedLoGe and FedETF:
>
> 1. FedETF is only designed to optimize a **single global** model, while our FedLoGe optimizes **both global model and client models** which is more appealing in practice.
> 2. The ETF classifier has been noted to **suffer from feature degeneration**, characterized by significant fluctuations between features and the class mean, while **FedLoGo alleviates this issue** by filtering out noisy smaller mean features, making features more concise and effective.
> 3. **FedLoGe is less computationally expensive and easier to adapt to existing algorithms.** DR loss is necessary for FedETF, which means some modifications are required when integrating with existing algorithms. Besides, FedETF needs a projection layer for optimization. For ResNet18 with four blocks, a 512x512 projection layer would be approximately 1MB, almost doubling the parameter size of Block 1, which is 0.56MB. We conducted an ablation study on DR Loss and the Projection Layer, as shown in **the table below**. The results demonstrate that FedLoGe can achieve good performance even without the DR Loss and Projection Layer.

---

> ### Author Response · Authors · 2023-11-17
> **Response for Reviewer Nmh4 [3/3]:**
>
> | Method  | DR Loss | Projection Layer | Many   | Medium | Few    | All    |
> | ------- | ------- | ---------------- | ------ | ------ | ------ | ------ |
> | CreFF   | -       | -                | 0.684  | 0.440  | 0.146  | 0.401  |
> | FedGraB | -       | -                | 0.683  | 0.553  | 0.221  | 0.411  |
> | FedETF  | ×       | ×                | 0.6904 | 0.4751 | 0.1820 | 0.3825 |
> | FedETF  | √       | ×                | 0.6882 | 0.4607 | 0.2059 | 0.3932 |
> | FedETF  | √       | √                | 0.6909 | 0.5011 | 0.2310 | 0.4109 |
> | FedLoGe | ×       | ×                | 0.7137 | 0.4989 | 0.2179 | 0.4249 |
>
> **Please refer to the detailed discussion of the differences between our work and previous work in the responses to Reviewer tSL5 (W4), or in Section A.11 of our paper.**
>
> > Q5. How to optimize Eq (4) that contains a maximization over j?
>
> Thank you for your question regarding the details of Equation (4). We select two classifier vectors $\mathbf{\hat{\mathbf{\psi}}^{\prime}}_{:,i}$
>
> and $\mathbf{\hat{\mathbf{\psi}}^{\prime}}_{:,j}$ $(i \neq j)$ which have maximum cosine similarity. Then we transfer cosine similarity to degree and optimize it. The calculation steps for Maximal Angle Loss are shown in the code below:
>
> ```python
> # Constraint 2: Maximize the angle between vectors (minimize cosine similarity)
> normalized_etf = sparse_etf / row_norms
> cos_sim = torch.mm(normalized_etf.t(), normalized_etf)
> torch.diagonal(cos_sim).fill_(-1)
> angle_loss = -torch.acos(cos_sim.max(dim=1)[0].clamp(-0.99999, 0.99999)).mean()
> ```
>
> We shall provide a more detailed explanation of the details of the optimization process.
>
> Please access more details about the Maximal Angle Loss in **Supplementary Material** ( ./util/etf_methods.py).
>
> Q6. What is the purpose of introducing sparsity to the SSE-C classifier, besides its empirical effectiveness?
> Thank you for your insightful inquiry about the role of sparsity in the SSE-C classifier. The introduction of sparsity serves several key purposes beyond its empirical effectiveness, as outlined below:
>
> 1. Non-sparse ETF results in unstable distances between features with smaller samples means and the class means, while enforcing **sparsity would mitigate feature degeneration** by squeezing poor-quality and noisy features into these sparsely allocated positions. We demosntrate this in **Fig. 1.** Please check Q3 for more details.
> 2. Introducing sparsity helps **learn more concise and dominant features** for better classification, which is evident in **Figures 3(b) and 3(c)**. It shows that after training with SSE-C, pruning dominant features drastically decreases performance, while pruning negligible features barely affects performance.
> 3. Sparsity can help **learn more discriminant decision boundaries**. We have conducted a T-SNE visualization of the class means before and after sparsification, detailed in **Appendix A.4**. This visualization reveals that the decision boundaries of class mean post-sparsification are more distinct.
>
> [1] Bbn: Bilateral-branch network with cumulative learning for long-tailed visual recognition. CVPR 2022.
>
> [2] Decoupling representation and classifier for long-tailed recognition. ICLR 2020.
>
> [3] Equalization loss v2: A new gradient balance approach for long-tailed object detection. CVPR 2021.
>
> [4] Over-coming classifier imbalance for long-tail object detection with balanced group softmax. CVPR 2020.
>
> [5] Adjusting decision boundary for class imbalanced learning. IEEE Access 2020.
>
> [6] Neural collapse in deep linear network: From balanced to imbalanced data. ICML 2023.
>
> [7] Imbalance trouble: Revisiting neural-collapse geometry. NIPS 2022.
>
> [8] Federated Learning on Heterogeneous and Long-Tailed Data via Classifier Re-Training with Federated Features. IJCAI 2022.
>
> [9] Fedic: Federated learning on non-iid and long-tailed data via calibrated distillation. IEEE ICME 2022.
>
> [10] Fed-GraB: Federated Long-tailed Learning with Self-Adjusting Gradient Balancer. NIPS 2023.
>
> [11] BalanceFL: Addressing Class Imbalance in Long-Tail Federated Learning. IEEE IPSN 2022.

---

> ### Author Response · Authors · 2023-11-23
> **Kind Reminder: Review Deadline Approaching**
>
> Dear Reviewer Nmh4:
>
> As the review deadline is nearing, with just five hours remaining, we wish to highlight our recent efforts in addressing your concerns. We've provided clear motivations for each algorithmic step, enhanced Fig. 1's clarity, conducted a thorough comparison with related work, and clarified intricate details of our algorithm design.
>
> We trust these clarifications will assist in your re-evaluation. Your feedback is invaluable, and we thank you for your time and attention.
>
> Warm regards,
>
> Authors

---

### Author Response · Authors · 2023-11-22
**General Response to All Reviewers**

Dear Reviewers:

Thank you again for your time and insightful comments! We have comprehensively revised our work according to your comments (please kindly refer to the rebuttal below). We hope we have addressed your concerns regarding the details of the setting, the novelty of the task and method, etc. **Since the discussion is about to close, we would be grateful if you would kindly let us know of any other concerns and if we could further assist in clarifying any other issues.**

Thanks a lot again, and with sincerest best wishes

Authors

---

### Meta-Review · Area_Chair_Lxb4 · 2023-12-07

**Metareview:**

This paper focuses on the federated long-tailed learning problem by using the neural collapse technique. Specifically, a static sparse equiangular tight frame classifier is proposed to enhance representation learning, and the Global and Local Adaptive Feature Realignment (GLA-FR) is used to align global features with client preferences. Extensive experiments also demonstrate the efficacy of the proposed approach.

Strengths:

(1)   This paper is well written. The long tail federated learning is an important problem that is worthy of studying.

(2)   The idea that uses neural collapse to solve the long tail federated learning is well-motivated.

(3)   The experiments are extensive to demonstrate the efficacy of the proposed approach. The proposed approach provides a new SOTA for federated long-tail federated learning.

Weaknesses:

(1)   The overall pipeline of the proposed framework follows the structure of FedRod, which also uses the shared feature extractor, shared classifier, and personalized classifier. The authors should point out the difference between this work and FedRod. The dual classifier structure credit should go to FedRod. It seems that this work is a combination of FedETF and FedROD. In the next version, the authors should mention the difference clearly to show the novelty of the proposed new method.

(2)   The writing of this work should largely improve. Too many notations are used without explanations. We recommend the authors summarize the notations in a Table to improve the readability.

After the authors' response and discussion with reviewers, most concerns are well solved. Although a few minor issues exist in the current version, the positive contributions of this work still beat weaknesses. Therefore, I recommend acceptance.

**Justification For Why Not Higher Score:**

N/A

**Justification For Why Not Lower Score:**

N/A

---

### Decision · Program_Chairs · 2024-01-16

Accept (poster)